# Patch-Prompt Aligned Bayesian Prompt Tuning for Vision-Language Models

Xinyang Liu[*1]    Dongsheng Wang[*1]    Bowei Fang[1]    Miaoge Li[1]    Yishi Xu[1]    Zhibin Duan[1]    Bo Chen[†1]

Mingyuan Zhou[2]

[1]National Key Laboratory of Radar Signal Processing, Xidian University, Xi'an, 710071, China.
[2]McCombs School of Business, The University of Texas at Austin, Austin, TX 78712, USA

## Abstract

For downstream applications of vision-language pre-trained models, there has been significant interest in constructing effective prompts. Existing works on prompt engineering, which either require laborious manual designs or optimize the prompt tuning as a point estimation problem, may fail to describe diverse characteristics of categories and limit their applications. We introduce a Bayesian probabilistic resolution to prompt tuning, where the label-specific stochastic prompts are generated hierarchically by first sampling a latent vector from an underlying distribution and then employing a lightweight generative model. Importantly, we semantically regularize the tuning process by minimizing the statistical distance between the visual patches and linguistic prompts, which pushes the stochastic label representations to faithfully capture diverse visual concepts, instead of overfitting the training categories. We evaluate the effectiveness of our approach on four tasks: few-shot image recognition, base-to-new generalization, dataset transfer learning, and domain shifts. Extensive results over 15 datasets show promising transferability and generalization performance of our proposed model, both quantitatively and qualitatively.

## 1 INTRODUCTION

Large-scale vision-language pre-trained models (VLPs) have recently demonstrated impressive achievements on various computer vision tasks [Wang et al., 2021, Jia et al., 2021, Cho et al., 2021, Radford et al., 2021, Li et al., 2022]. Pre-trained on web-scale image-text association pairs, such VLPs have the ability to carry the semantic knowledge on

which visual concepts correspond to which textual sequence and vice versa, and this has been proven beneficial for visual understanding [Radford et al., 2021, Mei et al., 2022, Du et al., 2022]. This has motivated the rapid rise of *prompt tuning* that hopes to fine-tune VLPs by formalizing the downstream tasks as language modeling problems and optimizing only the text inputs (prompts) [Radford et al., 2021, Zhou et al., 2022a,b], such as "*X X X X {class}*.", where "*X*" and "*{class}*" denotes the prefix tokens and real class names, respectively. In contrast to supervised learning with discrete labels from a closed set of categories, prompt tuning receives knowledge from pre-trained language models and supports open-set visual concepts, often producing better performance, especially on few/zero-shot tasks [Zhou et al., 2022a, Gu et al., 2022].

To specify the optimal prefix tokens "*X*" that provide rich context for pre-trained language models, prompt tuning methods often optimize them as learnable embedding vectors with a task-specific loss. For example, CoOp [Zhou et al., 2022b] employs the cross entropy loss to learn 16 prefix tokens that are shared across all categories and finds that such data-driven paradigms achieve significant improvement over hand-crafted prompts. However, recent studies report that the overfitting issue occurs in the training process and often leads to poor generalizability and transferability [Zhu et al., 2022, Ma et al., 2022, Lu et al., 2022]. To this end, various techniques are introduced under different assumptions, including conventional anti-overfitting tricks, instance-specific prompt generation, and gradient flow [Gao et al., 2021, Zhou et al., 2022a, Ma et al., 2022, Zhu et al., 2022]. Another concern stems from deterministic prompt learning, where the prompts are learned as the point estimation, and only a single sentence is searched to represent a given class. Intuitively, one class can be characterized by multiple intrinsic attributes (See Fig 1 for example). Thus, it is critical to learn multiple prompts that focus on different concepts. Motivated by this, several previous works attempt to learn multiple prompt [Chen et al., 2022] or introduce distributed prompt embeddings [Derakhshani et al.,

---

[*]Equal contribution: {xinyangatk, wds}@stu.xidian.edu.cn
[†]Corresponding author: bchen@mail.xidian.edu.cn

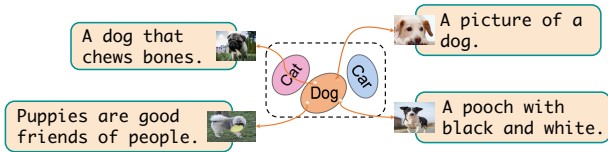

Figure 1: The motivation of the proposed model. Multiple prompts are generated from the label-specific distributions.

2022, Lu et al., 2022, Wang et al., 2023], showing a large improving gap over the baseline method. However, those models either require pre-defined prompts or focus on the sample-dependent prompt generation, failing to discover label-specific prompts efficiently.

To address the above shortcomings, we in this paper propose Bayesian prompt tuning, where label-specific stochastic prompts are generated hierarchically under the Bayesian framework. As illustrated in Fig 1, one of the core ideas is to generate multiple prompts for the given categories, with each of the learned prompt capturing various visual attributes, resulting in diverse and generalizable prompt discovery. Specifically, we first introduce uncertainty in the latent embedding space and model each category as a variational distribution [Kingma and Welling, 2014]. Compared to previous point estimation methods, this approach enables us to infer a posterior distribution that contains meta-information about the corresponding category, offering advantages in modeling uncertainty and highly structured data [Fan et al., 2020]. To complete the prompt sentence, a sequence generation module is employed to generate the prefix sequence according to the meta-vector sampled from the underlying distribution. Note that various language models can be chosen as the generator, e.g., the LSTM [Hochreiter and Schmidhuber, 1997] and transformers [Al-Rfou et al., 2019]. Although the generator itself is a deterministic mapping, the output prompts can be viewed as an implicit distribution in the embedding space due to its stochastic inputs. This property allows our proposed model to naturally handle diverse visual concepts, resulting in robust prompt tuning.

Furthermore, to tackle the issue of overfitting in prompt tuning, we propose a novel semantic regularization approach that leverages the conditional transport (CT) framework [Zheng and Zhou, 2021] to establish a relationship between visual patches and textual prompts. Specifically, we use the modality-specific outputs of CLIP to construct a visual patch set as well as a textual prompt set for each target image. The former is obtained by collecting the image patch embeddings and the latter is constructed from all label embeddings. Due to the shared common embedding space of CLIP, these two sets can be regarded as two discrete distributions over the same semantic space. They represent similar meanings about the target image, while from different modalities. Therefore, prompt tuning can be viewed as the process of learning the distribution of textual prompts

to be as close to the distribution of visual patches as possible. Fortunately, the recent developments in CT provide us with an efficient tool to quantify the difference between two discrete distributions [Tanwisuth et al., 2021, Wang et al., 2022, Tanwisuth et al., 2023]. Importantly, the distance function in CT specifies the similarities between the prompt embeddings and visual patches in the embedding space, which makes it possible to regularize the learning of prompts with visual guidance. As a result, the aligned prompts are encouraged to capture the true label-specific visual concepts, rather than over-fitting to the training set.

The main contributions of this paper are summarized as follows:

- We propose Bayesian prompt tuning that generates label-specific stochastic prompts hierarchically, models each label as a distribution over the embedding space and successfully handles diverse visual concepts.

- To avoid over-fitting to the training set, we introduce the CT distance as a regularization that guides the learning of prompts with visual knowledge by aligning the patches and prompt embeddings semantically.

- We formulate the proposed model as a variational inference problem, and a combined loss function is derived to optimize all parameters efficiently. Extensive experiments show that our models outperform the baselines.

## 2 THE PROPOSED METHOD

An overview of our proposed **P**atch-prompt aligned **B**ayesian prompt tuning (PBPrompt) is shown in Fig 2. Below, we first briefly review CoOp, which is the basic concept used in this paper. Then, we introduce the details of our model, which aims to improve the diversity and generalizability of CoOp.

### 2.1 REVIEWS OF COOP

Context Optimization (CoOp) [Zhou et al., 2022b] is built on CLIP-like VLPs and is a pioneering method for continuous prompt tuning. A VLP often consists of an image encoder $f$ and a text encoder $g$, each taking modality-specific sequence as inputs and outputs $d$-dimensional vectors in the shared embedding space. Prompt tuning methods usually design a template to construct the category descriptions and then view the outputs of $g$ as the class weight for the classification task. To address the limitation of handcrafted templates and facilitate the learning of optimal prompts for adapting VPLs to downstream tasks, CoOp models each prompt token as a continuous vector that can be learned from data. *E.g.*, the prompt for $c$-th class can be denoted as: $\boldsymbol{t}_c = [\boldsymbol{v}_1, \boldsymbol{v}_2, ..., \boldsymbol{v}_b, \boldsymbol{e}_c]$, where $\boldsymbol{e}_c$ is the label embedding of class $c$, $\boldsymbol{v} = \{\boldsymbol{v}_i \in \mathbb{R}^d\}_{i=1}^b$ are $b$ learnable context vectors. Given a set of category descriptions $\{\boldsymbol{t}_c\}_{c=1}^C$ and an image

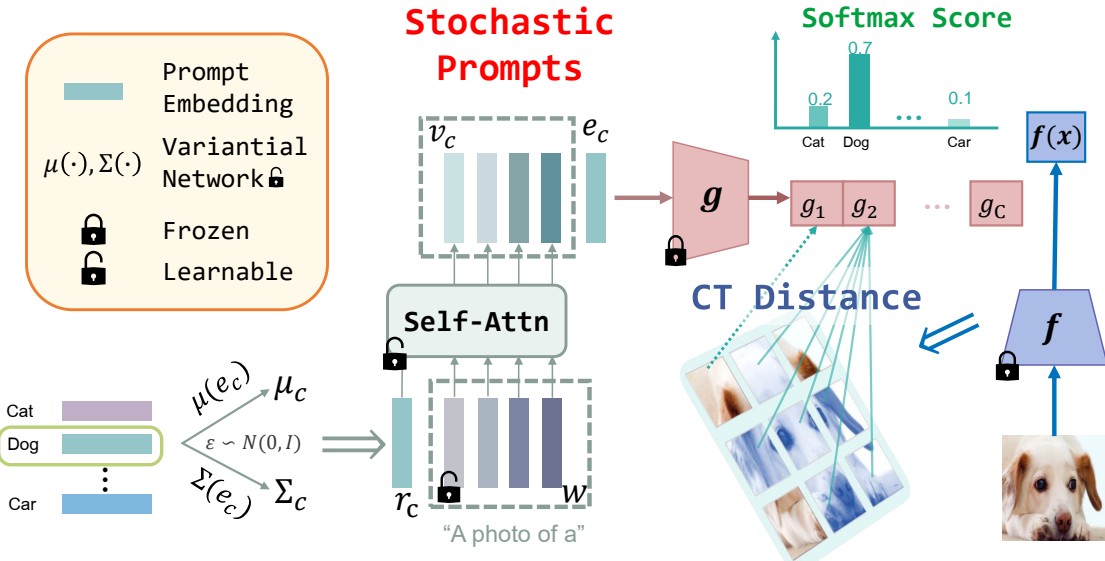

Figure 2: Overview of the proposed PBPrompt. PBPrompt generates the stochastic prompts by first sampling a label-specific vector $r_c$ and then employing a single-layer self-attention generator. CT distance is performed between the textual prompts and image patches to regularize the prompts with the visual knowledge.

$x \in \mathbb{R}^{(3 \times H \times W)}$, CoOp models the image label $p(y|x)$ as a categorical distribution according to the similarity between the image and label features with:

$$p(y = c|x) = \frac{\exp(\text{sim}(f(x), g(t_c))/\tau)}{\sum_{c'}^{C} \exp(\text{sim}(f(x), g(t_{c'})/\tau)}, \quad (1)$$

where $\text{sim}(\cdot, \cdot)$ means the similarity function, e.g., the cosine similarity, and $\tau$ is the temperature parameter. Then one can optimize the prefix embeddings $v$ by back-propagating the following loss through the frozen VLPs with a few training samples $\mathcal{D}^{\text{tr}} = \{(x_i, y_i)\}_{i=1}^{N_{tr}}$:

$$\mathcal{L}(v) = \mathbb{E}_{x_i, y_i}[-\log p(y_i \mid x_i; v)]. \quad (2)$$

After tuning, $t_c$ can be used to define the target classifier for open-set image classification.

## 2.2 PATCH-PROMPT ALIGNED BAYESIAN PROMPT TUNING

The core idea behind the proposed PBPrompt is to learn distributed label-specific prompts under the Bayesian framework, as well as align the image patches and textual prompts by minimizing the CT distance. Below, we introduce the details of PBPrompt, which consists of stochastic prompt generation, patch-prompt alignment, and the training algorithm.

**Stochastic Prompts Generation (SPG)**   Generally, it is less sound to represent one class with a deterministic point, which may fail to cover diverse visual concepts, e.g., the object type, size, color, and so on. This issue becomes particularly acute in cases involving distribution shifts. For

instance, a model may see an image of a dog playing on the green ground during training but fail to make a correct prediction of another image of a dog on the beach. To this end, one of the goals of PBPrompt is to introduce uncertainty into prompt generation. For a target label, we assume there are various prompts that can achieve similar performance. These prompts originate from the same target class but depict its representative attributes from different perspectives, resulting in robust representation. An intuitive approach is to model the prompts as a distribution $p(r)$. Unfortunately, directly learning such a distribution over a sequence of $b$ vectors feature dimension $d$ is not simple [Brown et al., 2020, Lu et al., 2022], especially under the few-shot setting. To this end, we move the uncertainty forward to its inputs and develop a hierarchical generative module to produce the stochastic prompts:

$$t_c = [\phi(v_c \mid r_c), e_c], \quad r_c \sim p(r_c), \quad (3)$$

where $p(r_c)$ denotes the label-specific distribution that handles the conceptual diversity of class $c$. $\phi(v_c \mid r_c)$ denotes the deterministic generative model that takes the sampled $r_c$ as input and outputs the prefix token sequence $v_c = \{v_{c,l} \in \mathbb{R}^d\}_{l=1}^b$. Like previous works [Zhou et al., 2022b,a], the final prompt input $t_c$ is obtained by adding the label embedding $e_c$ at the end of prefix tokens. Different from previous models that view $t_c$ as the learnable embedding vectors, we generate $t_c$ via a hierarchical path, where a stochastic vector $r_c$ is first sampled from the label-specific distribution and the prefix sequence $v_c$ is then generated according to $r_c$. Although the generative model $\phi$ is a deterministic network, $t_c$ can be viewed as an implicit distribution over $r_c$. In this way, multiple prompts can be generated by sampling various $r_c$.

Note that $\phi(\boldsymbol{v}_c \mid \boldsymbol{r}_c)$ can be implemented with various language models Greff et al. [2017], Devlin et al. [2019], and we find a single-layer self-attention network works well in most cases [Vaswani et al., 2017], empirically:

$$
\begin{aligned}
\boldsymbol{s}_c &= [\boldsymbol{r}_c + \text{PE}_1, \boldsymbol{w}_1 + \text{PE}_2, ..., \boldsymbol{w}_b + \text{PE}_{b+1}], \\
[\hat{\boldsymbol{r}}_c, \boldsymbol{v}_{c,1}, ..., \boldsymbol{v}_{c,b}] &= \phi(\boldsymbol{v}_c|\boldsymbol{r}_c) := \text{Self-Attn}(\boldsymbol{s}_c),
\end{aligned}
\quad (4)
$$

where $\boldsymbol{w} = [\boldsymbol{w}_1, ..., \boldsymbol{w}_b]$ is the initialized prefix embeddings, and PE is the learnable position embedding matrix that captures the sequential relations of prefix tokens. The Self-Attn decoder takes $\boldsymbol{s}_c$ as inputs, where the sampled $\boldsymbol{r}_c$ in Eq. 3 is viewed as a special label token presented at the beginning of the initialized prefix sequence. It then outputs the class-specific prefix sequence $\boldsymbol{v}_c = [\hat{\boldsymbol{r}}_c, \boldsymbol{v}_{c,1}, ..., \boldsymbol{v}_{c,b}]$. This process allows the output tokens to encompass both contextual information and class-specific guidance, resulting in the generation of meaningful prompts.

**Regularization Between Textual Prompts and Visual Patches** Notably, the core motivation behind SPG is to learn diverse prompts that cover multiple visual concepts. However, directly optimizing SPG with the classification loss may suffer from the mode-collapse problem, where the sampled $\boldsymbol{r}_c$ tends to be close to each other, leading to single-mode prompt tuning. *E.g.*, the learned prompt pattern overfits the training set while failing to provide the true context. To address this issue, we introduce the regularization between the prompt outputs and image patches. This regularization encourages the sampled prompts to be close to a variety of patch embeddings, preventing them from overfitting to the training mode.

Recall that a VLP describes target labels from both the image and text domains. The former divides an image $\boldsymbol{x}$ into $M$ patches $\boldsymbol{u} = \{\boldsymbol{u}_m|_{m=1}^M\} \in \mathbb{R}^{d \times M}$, which provides the local visual features. We view the output embeddings of the textual encoder as the class-specific features, which provide the linguistic description for classes. Mathematically, given $\boldsymbol{x}$ and its prediction probability $\boldsymbol{p} = p(\boldsymbol{y}|\boldsymbol{x})$, we formulate those two sets as discrete distributions:

$$
P = \sum_{m=1}^M \frac{1}{M} \delta_{\boldsymbol{u}_m}, \quad Q = \sum_{c=1}^C p_c \delta_{\boldsymbol{g}_c} \quad (5)
$$

where $\delta$ is the Dirac delta function, $\boldsymbol{g}_c = g(\boldsymbol{t}_c)$ is the textual outputs of label $c$. Eq. 5 represents $\boldsymbol{x}$ as a mixture of patch embeddings $P$ and a mixture of prompt embeddings $Q$, both sharing the same semantics but originating from different domains. Naturally, we aim to regularize the learning of $Q$ by aligning it to $P$. A common choice is to minimize the optimal transport (OT) between $P$ and $Q$ [Cuturi, 2013, Chen et al., 2022]. However, the calculating of OT struggles in two-stage iterations: first solving for the transport plan and then updating the network, leading to unstable training. Fortunately, the recently developed conditional transport (CT) [Zheng and Zhou, 2021] offers an efficient tool to

align two distributions over different supports [Wang et al., 2022, Tanwisuth et al., 2021]. The CT distance between the textual prompts and visual patches is defined from two directions:

$$
\mathcal{L}_{CT}(P, Q) = \mathcal{L}_{\boldsymbol{u} \to \boldsymbol{g}} + \mathcal{L}_{\boldsymbol{g} \to \boldsymbol{u}}, \quad (6)
$$

where $\mathcal{L}_{\boldsymbol{u} \to \boldsymbol{g}}$ denotes the transport distance from patch embeddings to prompts, while $\mathcal{L}_{\boldsymbol{g} \to \boldsymbol{u}}$ denotes the transport distance in the reverse direction. The transport distance from patch embeddings to prompts can be calculated as:

$$
\mathcal{L}_{\boldsymbol{u} \to \boldsymbol{g}} = \frac{1}{M} \sum_{m=1}^M \sum_{c=1}^C \mathcal{C}(\boldsymbol{u}_m, \boldsymbol{g}_c) \pi(\boldsymbol{g}_c|\boldsymbol{u}_m), \quad (7)
$$

where $\mathcal{C}(\boldsymbol{u}_m, \boldsymbol{g}_c)$ is the cost function that measures the point-wise transport cost from $m$-th patch to $c$-th prompt embedding, *e.g.*, $\mathcal{C}(\boldsymbol{u}_m, \boldsymbol{g}_c) = 1 - cosine(\boldsymbol{u}_m, \boldsymbol{g}_c)$. $\pi(\boldsymbol{g}_c|\boldsymbol{u}_m) = \frac{p_c \exp(\boldsymbol{u}_m^T \boldsymbol{g}_c)}{\sum_{c'=1}^C p_{c'} \exp(\boldsymbol{u}_m^T \boldsymbol{g}_{c'})}$ is the transport plan. The core idea of Eq. 7 is to assign $M$ patches to their expected prompts. This can be viewed as a clustering process that learns a semantic center for each class-specific prompt. Unfortunately, only with $\mathcal{L}_{\boldsymbol{u} \to \boldsymbol{g}}$, many less-related patches within an image may be assigned to the target prompt. This may push the stochastic prompt to an average point, leading to mode collapse. To address this issue, CT introduces $\mathcal{L}_{\boldsymbol{g} \to \boldsymbol{u}}$ from an opposite direction:

$$
\mathcal{L}_{\boldsymbol{g} \to \boldsymbol{u}} = \sum_{c=1}^C p_c \sum_{m=1}^M \mathcal{C}(\boldsymbol{g}_c, \boldsymbol{u}_m) \phi(\boldsymbol{u}_m|\boldsymbol{g}_c), \quad (8)
$$

where $\pi(\boldsymbol{u}_m|\boldsymbol{g}_c) = \frac{\exp(\boldsymbol{g}_c^T \boldsymbol{u}_m)}{\sum_{m'=1}^M \exp(\boldsymbol{g}_c^T \boldsymbol{u}_{m'})}$. Unlike $\mathcal{L}_{\boldsymbol{u} \to \boldsymbol{g}}$ which has the patch-clustering effect, $\mathcal{L}_{\boldsymbol{g} \to \boldsymbol{u}}$ aims to push the expected prompt towards patches that semantically close to it, creating a prompt-covering effect. The CT distance in Eq. 6 provides us with a novel regularization, enabling the learning of stochastic prompts with vision knowledge from bi-directions. The *patch-to-prompt* transportation explores meaningful prompt outputs, and the *prompt-to-patch* transportation improves the uncertainty of the prompt outputs.

## 2.3 TRAINING WITH COMBINED ELBO

Given the VLPs and labeled images $\mathcal{D}^{\text{tr}}$, we would like to distill the pre-trained knowledge and learn the posterior of the label-specific representation $p(\boldsymbol{r}_c|\mathcal{D}^{\text{tr}})$ as well as the deterministic generative model $\phi(\boldsymbol{v}_c|\boldsymbol{r}_c)$. Unfortunately, the exact posterior for $\boldsymbol{r}_c$ is intractable and needs to be approximated. To this end, we define the variational distribution $q(\boldsymbol{r}_c|c)$ and employ the variational inference to optimize the proposed method by minimizing the following combined Evidence Lower BOund (ELBO) [Kingma and Welling, 2014]:

$$
\begin{aligned}
\mathcal{L} = &-\mathbb{E}_{\boldsymbol{t}_c=[\pi(\boldsymbol{v}_c|\boldsymbol{r}_c), \boldsymbol{e}_c], \boldsymbol{r}_c \sim q(\boldsymbol{r}_c|c)} \log p(y|\boldsymbol{x}, \boldsymbol{t}_c) \\
&- \text{D}_{\text{KL}}[q(\boldsymbol{r}_c|c)||p(\boldsymbol{r}_c)] + \eta \mathcal{L}_{CT}(P, Q),
\end{aligned}
\quad (9)
$$

where we follow previous practices [Gordon et al., 2019, Derakhshani et al., 2022] and define the variational distribution $q$ as a Gaussian distribution conditioned on the label embedding $e_c$: $q(r_c|c) = \mathcal{N}(\mu(e_c), \Sigma(e_c))$, with $\mu$ and $\Sigma$ parameterized by two fully-connected layers. The first term in Eq. 9 is the expected log-likelihood defined at Eq.1, the second term is the KL-divergence that encourages the variational posterior to approach to its prior, and the last term is the CT distance that aligns the class-specific prompt with image patches. $\eta$ denotes the trade-off hyperparameter that controls the regularization weights. Unlike most previous works that solely learn prompts from task-specific loss [Zhou et al., 2022b, Lu et al., 2022], we optimize the proposed PBPrompt with combined ELBO that introduces the CT distance as a regularization, guiding the label embeddings to focus on meaningful visual concepts rather than over-fitting to the base sets. We summarize the training algorithm at the Algorithm 1 in Appendix.

**Contextual Prior** $p(r_c)$    Instead of treating the prior as a fixed distribution independent of the label $c$, here we define the label-specific priors to further explore label semantics via the label embeddings, *e.g.*, $p(r_c) = \mathcal{N}(e_c, I)$. Thus compared to the fixed prior, the proposed label-specific prior introduces additional label semantics and achieves better prior guidance.

## 3   RELATED WORK

The technique of prompt tuning, originating from the natural language processing (NLP) domain and aims at best utilizing pre-trained language models [Brown et al., 2020, Shin et al., 2020, Liu et al., 2023], has gained increasing research attention in VLPs due to its impressive results [Ge et al., 2022, Sun et al., 2022, Feng et al., 2022]. For example, CLIP [Radford et al., 2021] manually designs templates based on human knowledge and shows great potential in few/zero-shot tasks. Context Optimization (CoOp) [Zhou et al., 2022b] first introduces the continuous prompt into VLPs and views the prompt tokens as a set of learnable vectors that can be optimized by minimizing the cross entropy loss. Instead of learning static prompts, Conditional CoOp (CoCoOp) [Zhou et al., 2022a] learns an input-specific prompt by incorporating image features via a lightweight network and shows better generalization on unseen categories. The most related work to ours is distributed prompt tuning, which focuses on stochastic prompt tuning. For instance, Prompt Distribution leArning (ProDA) [Lu et al., 2022] first designs multiple handcrafted templates and then employs a Gaussian distribution to model the latent representation. Variational prompt tuning (VPT) of [Derakhshani et al., 2022] constructs prompt tokens by directly adding Gaussian samples into prompt vectors. SyntHesIzed Prompt (SHIP) of [Wang et al., 2023] samples a image-dependent prompt by training a VAE with the image features. Prompt learning with optimal transport (PLOT) [Chen et al., 2022]

applies optimal transport theory to learn multiple local prompts. While all above methods—ProDA, VPT, and SHIP, PLOT, and ours—involve learning stochastic prompts, they are fundamentally distinct. We model each target label as a Gaussian distribution and then generate stochastic prompts based on label-specific samples, resulting in better label representations.

## 4   EXPERIMENTS

We follow the exact experimental setup of previous works [Zhou et al., 2022b,a] and validate the performance of PBPrompt against the recent state-of-the-art prompt learning models on widely-used benchmarks under various settings, including few-shot learning, base-to-new generalization, cross-dataset transferability, and domain generalization.

### 4.1   EXPERIMENTAL SETUP

**Datasets.** For the first two tasks, we rely on 11 classification datasets, *i.e.*, ImageNet [Deng et al., 2009] and Caltech101 [Fei-Fei et al., 2004] for generic object classification, OxfordPets [Parkhi et al., 2012], Stanford-Cars [Krause et al., 2013], Flowers102 [Nilsback and Zisserman, 2008], Food101 [Bossard et al., 2014] and FGVCAircraft [Maji et al., 2013] for fine-grained image recognition, EuroSAT [Helber et al., 2019] for satellite image classification, UCF101 [Soomro et al., 2012] for action classification, DTD [Cimpoi et al., 2014] for texture classification, and SUN397 [Xiao et al., 2010] for scene recognition. For the domain generalization task, we use ImageNet as the source domain dataset and evaluate performance on ImageNetV2 [Recht et al., 2019], ImageNet-Sketch [Wang et al., 2019], ImageNet-A [Hendrycks et al., 2021b], and ImageNet-R [Hendrycks et al., 2021a]. The details of each dataset are provided at Table C. 1.

**Baselines.** We compare our proposed approach with following state-of-the-art (SoTa) models: zero-shot CLIP [Radford et al., 2021] with the fixed handcrafted prompt *"A photo of a {class}."*, CoOp [Zhou et al., 2022b], CoCoOp [Zhou et al., 2022a], PLOT [Chen et al., 2022], and stochastic prompt tuning methods, including ProDA [Lu et al., 2022], VPT [Derakhshani et al., 2022] and SHIP [Wang et al., 2023].

**Implementation Details.** Similar to previous works [Zhou et al., 2022b,a], PBPrompt adopts the vision and language encoders as a ViT-B/16 [Dosovitskiy et al., 2020] and transformer [Vaswani et al., 2017] respectively. We consistently perform prompt tuning with 16 shots and fix the prompt length as 4 for the four primary image classification tasks across all datasets. We set the trade-off hyperparameter $\eta$ as 0.01 and run each experiment with 10 epochs on base-to-new generalization. The label embedding $e_c$ is obtained

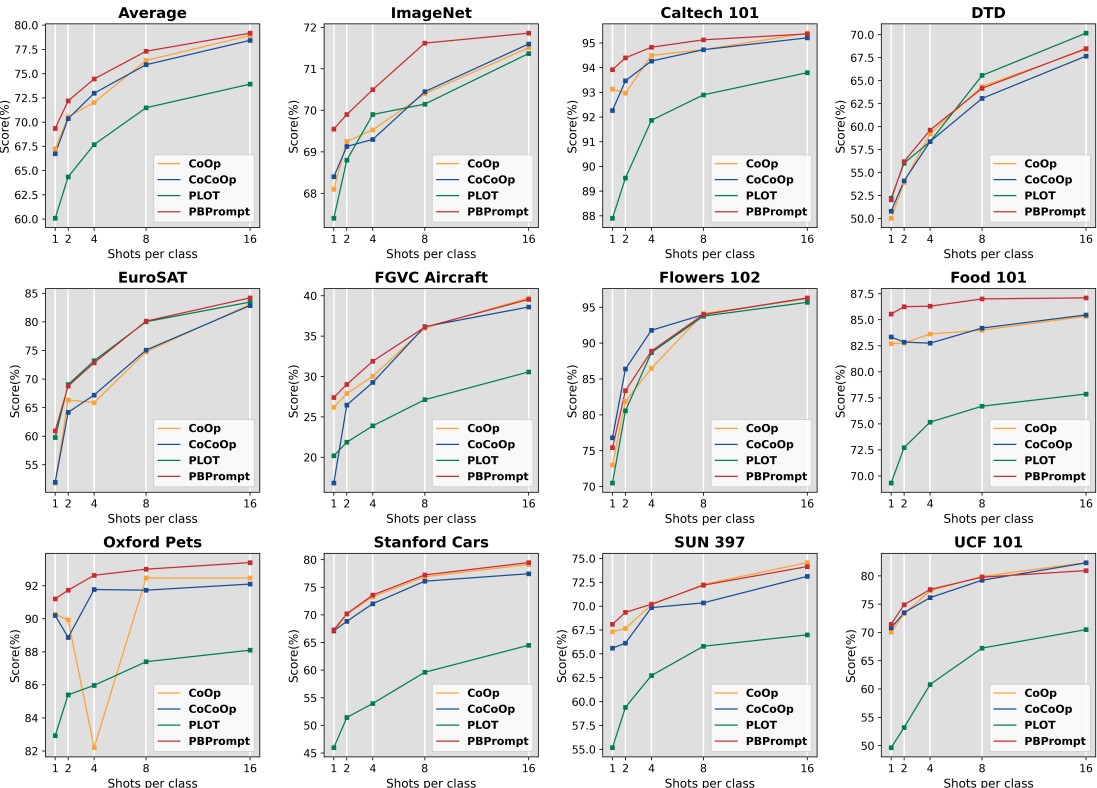

Figure 3: The few-shot learning results on 11 datasets. We compare our PBPrompt with CoOp, CoCoOp and PLOT. Overall, our proposed model outperforms the baselines in most cases. More numerical results can be found at Table C. 5 and Table C. 6.

by averaging the CLIP embedding of the class names, and we initialize the learnable prompt embedding vectors from $\mathcal{N}(0, 0.02)$. For the self-attention network in (4), we employ 8 heads for deeper interactions between prompt tokens. We summarize the training details in the appendix. The results for CoOp and CoCoOp are adopted The results for CoOp and CoCoOp are adopted from the published papers, except for the few-shot learning experiments. For these experiments, we re-ran them using the same settings, with a maximum epoch set to 200 for 16/8 shots, 100 for 4/2 shots, and 50 for 1 shot across all datasets. For a fair comparison, we re-run PLOT with ViT-B/16 on all the experiments in the settings above. All results are reported as the mean value over three seeds.

## 4.2 EXPERIMENT RESULTS

**Few-shot Learning** evaluates a model's capability to handle limited labeled data and samples. The complete results are summarized in Fig 3, where we find that 1) our method consistently outperforms the baseline models across various scenarios, and 2) PBPrompt outperforms other methods when trained with 1, 2, and 4 shots, showcasing a substantial performance margin on DTD, EuroSAT, Flowers102, and FOOD101 datasets. Furthermore, as the number of training samples increases, the performance gap between models diminishes, particularly evident in the case of training with

8/16 shots. This emphasizes the exceptional performance of our model in few-shot learning tasks. Notably, PBPrompt surpasses CoOp with average accuracy increases of 3.14%, 2.32%, 6.33%, 1.24%, and 0.32% at 1, 2, 4, 8, and 16 shots, respectively.

**Base-to-New Generalization** assesses model's generalizability in a zero-shot setting. We report the Base-to-New results at Fig 4 (The detailed accuracy on base and new set can be found at Table C. 8). Note that the H score is calculated as H = $(2 \times \text{Base} \times \text{New})/(\text{Base} + \text{New})$, which is a trade-off metric between the base and new sets. We find that PBPrompt surpasses other stochastic baselines in terms of H score across all datasets. This demonstrates the efficiency of the introduced label-specific SPG. Besides, due to the CT regularization, our approach successfully mitigates the overfitting issue, showing robust ability to balance the Base and New performance.

**Cross-Dataset Transfer Learning** measures the transfer performance from different sources, where we train our model on ImageNet (source dataset) and then test it on 10 distinct target datasets. As shown at Table 1, PBPrompt has improvements on 9 out of 10 target domains compared to CoCoOp, This demonstrates that the proposed PBPrompt has the potential to transfer from a single dataset. Moreover, we also find that PBPrompt exhibits large gaps on

| Method | Source Imagenet | Target Caltech | Pets | Cars | Flowers | Food | Aircraft | SUN | DTD | EuroSAT | UCF | Average |
|---|---|---|---|---|---|---|---|---|---|---|---|---|
| CoOp | 71.51 | 93.70 | 89.14 | 65.41 | 68.71 | 85.30 | 18.47 | 64.15 | 41.92 | 46.39 | 66.55 | 63.81 |
| CoCoOp | 71.02 | 94.43 | 90.14 | 65.32 | 71.88 | 86.06 | 22.94 | 67.36 | **45.73** | 45.37 | 68.21 | 65.74 |
| PBPrompt | **71.71** | **94.87** | **90.62** | **66.00** | **72.44** | **86.34** | **24.82** | **67.69** | 45.62 | **47.13** | **68.83** | **66.40** |
| Δ | +0.69 | +0.44 | +0.48 | +0.68 | +0.56 | +0.28 | +2.90 | +0.33 | −0.11 | +1.76 | +0.62 | +0.66 |

Table 1: Cross-dataset transfer learning accuracy results of various baselines on source and target datasets. Δ: The improvements of the proposed model compared to CoCoOp.

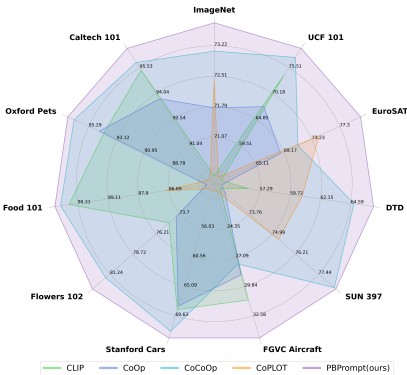

Figure 4: Performance comparison on base-to-new generalization evaluated by harmonic mean. More results can be found at Table C. 8 and C. 9.

| Method | Learnable | Source ImageNet | Target ImageNetV2 | ImageNet-Sketch | ImageNet-A | ImageNet-R |
|---|---|---|---|---|---|---|
| CLIP | ✗ | 66.73 | 60.83 | 46.15 | 47.77 | 73.96 |
| CoOp | ✓ | 71.51 | 64.20 | 47.99 | 49.71 | 75.21 |
| CoCoOp | ✓ | 71.02 | 64.07 | 48.75 | 50.63 | 76.18 |
| PBPrompt | ✓ | **71.71** | **64.53** | **49.32** | **51.64** | **76.71** |

Table 2: Cross-domain generalization accuracy results of various baselines.

| Backbones | | ViT-B/16 | | | RN50 | | |
|---|---|---|---|---|---|---|---|
| Dataset | | 1 shot | 2 shots | 4 shots | 1 shot | 2 shots | 4 shots |
| Caltech101 | CoOp | 93.19 | 92.97 | 94.50 | 87.51 | 87.84 | 89.52 |
| | PLOT | 87.90 | 89.53 | 91.87 | 89.83 | 90.67 | 90.80 |
| | B-Prompt | 93.57 | 94.10 | 94.75 | 90.10 | 89.70 | 90.56 |
| | P-Prompt | 93.34 | 93.95 | 94.60 | 88.54 | 89.45 | 90.70 |
| | PBPrompt | **93.92** | **94.40** | **94.83** | **90.21** | **90.86** | **90.92** |
| DTD | CoOp | 50.03 | 53.93 | 59.23 | 43.62 | 45.35 | 53.94 |
| | PLOT | **52.20** | **56.03** | 58.37 | 46.55 | 51.24 | 56.03 |
| | B-Prompt | 51.87 | 55.85 | 59.53 | 46.00 | 51.67 | 56.17 |
| | P-Prompt | 50.95 | 55.10 | 59.02 | 46.95 | 48.35 | 55.89 |
| | PBPrompt | 52.03 | 56.20 | **59.63** | **47.21** | **52.08** | **56.97** |
| FOOD101 | CoOp | 82.70 | 82.77 | 83.63 | 74.25 | 72.61 | 74.49 |
| | PLOT | 69.33 | 72.73 | 75.17 | **77.74** | 77.70 | 77.21 |
| | B-Prompt | 84.97 | 86.03 | 86.21 | 77.02 | 76.45 | 77.58 |
| | P-Prompt | 85.00 | 83.67 | 84.39 | 76.20 | 75.39 | 76.45 |
| | PBPrompt | **85.55** | **86.25** | **86.30** | 77.35 | **77.83** | **78.09** |
| SUN397 | CoOp | 67.32 | 67.67 | 70.14 | 60.12 | 59.60 | 63.24 |
| | PLOT | 55.17 | 59.40 | 62.73 | 62.47 | 61.71 | **65.09** |
| | B-Prompt | 67.98 | 69.00 | 70.20 | 62.42 | 63.03 | 64.83 |
| | P-Prompt | 67.45 | 68.25 | 70.10 | 62.10 | 61.54 | 64.12 |
| | PBPrompt | **68.10** | **69.35** | **70.21** | **62.51** | **63.45** | 64.77 |

Table 3: Ablation studies of backbones on few-shot learning.

fine-grained datasets (FGCVAircraft, OxfordPets, and Flowers102), suggesting the capacity to handle the discriminative features of each category.

**Domain Generalization** concerns about the robustness of the distribution shift, where we assess the proposed models on ImageNetV2, ImageNet-Sketch, ImageNet-A, and ImageNet-R after training it on the source dataset (ImageNet). We report the results at Table 2 and find that PBPrompt achieves the highest accuracy on all target domains compared to other baselines. This indicates that the learnable stochastic prompts are less sensitive to distribution shifts and can generalize well across domains.

### 4.3 FURTHER ANALYSIS

**Robustness and Synergistic Effect** In our previous experiments, we utilized the ViT-B/16 backbone. However, in this study, we also employ the RN50 backbone to assess the robustness of our model across different backbones. The few-shot learning accuracy results are presented in Table 3. As

demonstrated in the results, PBPrompt provides more consistent results than the prior state-of-the-art methods on both backbones, especially with the ViT-B/16 backbone, where PLOT suffers a significant performance drop in comparison. Additionally, we have compared two variants of PBPrompt, namely B-Prompt and P-Prompt, in few-shot learning and base-to-new tasks. B-Prompt contains only the SPG module, while P-Prompt only utilizes the conditional transport framework, both based on CoOp. We report the accuracy scores at Table 3 and Table 4 respectively. We observe that both variants exhibit significant improvements compared to CoOp, especially B-Prompt, which outperforms the previous methods in most of the test cases. Furthermore, PBPrompt achieves the highest performance on the majority of test cases among all methods by incorporating both variations, demonstrating the powerful synergistic effect of our approach.

**The effect of Monte Carlo sampling and $\eta$** Generally, increasing the number of samples in Monte Carlo sampling leads to stable results, but an appropriate number can introduce a moderate level of uncertainty, ultimately enhancing the model's generalization and representation capabilities.

Meanwhile, the hyperparameter $\eta$, which balances the regularization weights, plays a crucial role in establishing the connection between the stochastically generated prompts

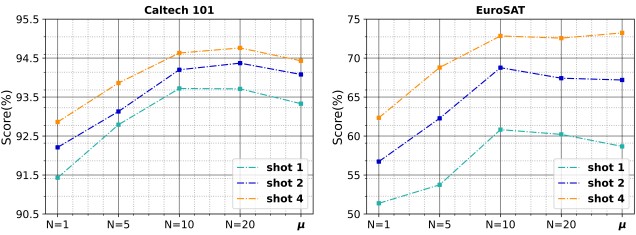

Figure 5: Monte Carlo sampling numbers

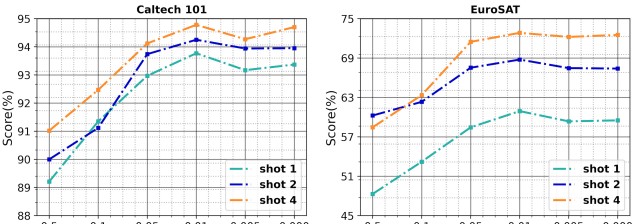

Figure 6: Regularization coefficient $\eta$

| Dataset | | CoCoOp | ProDA | VPT | SHIP | B-Prompt | P-Prompt | PBPrompt |
|---|---|---|---|---|---|---|---|---|
| **Caltech101** | Base | 97.96 | **98.27** | 95.47 | 97.55 | 97.35 | 97.95 | 97.98 |
| | New | 93.81 | 93.23 | 93.80 | 95.20 | 95.00 | 93.12 | **95.54** |
| | H | 95.84 | 95.68 | 94.62 | 96.36 | 96.16 | 95.47 | **96.74** |
| **Flowers102** | Base | 94.87 | **97.70** | 92.97 | 94.02 | 95.21 | 97.35 | 95.47 |
| | New | 71.75 | 68.68 | **75.90** | 74.40 | 72.35 | 69.57 | 73.60 |
| | H | 81.87 | 80.66 | 74.40 | 83.06 | 82.22 | 81.15 | **83.12** |
| **DTD** | Base | 77.01 | **80.67** | 57.67 | 74.88 | 77.20 | 79.97 | 78.03 |
| | New | 56.00 | 56.48 | **58.70** | 56.88 | 57.00 | 47.67 | 57.81 |
| | H | 64.85 | 66.44 | 58.18 | 64.65 | 65.58 | 59.73 | **66.42** |
| **EuroSAT** | Base | 87.49 | 83.90 | 67.97 | 88.62 | 87.21 | **92.46** | 89.53 |
| | New | 60.04 | 66.00 | 71.63 | 66.87 | 72.33 | 62.58 | **72.87** |
| | H | 71.21 | 73.88 | 69.75 | 76.22 | 79.08 | 74.64 | **80.35** |

Table 4: Base-to-New generalization results of various baselines. B-Prompt: Bayesian prompt tuning. P-Prompt: Patch-Prompt CT alignment. More resutls can be found at Table C.8.

and various visual concepts. We ablate these two hyperparameters on few-shot learning with 1/2/4 shots at Fig 5 and Fig 6. In Fig 5, we use $\mu$ to represent the simple adoption of the mean of multiple prompt embedding, and we observe that employing fewer samples leads to increased uncertainty and a significant drop in performance. This indicates that a higher number of samples is essential for achieving more reliable results. Fig 6 demonstrates that the presence of large coefficients can detrimentally impact results by overemphasizing image relationships, thus potentially overshadowing CLIP's alignment properties. We set the sampling number as 20 and $\eta = 0.01$ by default.

**Further ablation study** Due to space constraints, details of other interesting ablation study can be found in the Appendix and now they are briefly introduced as follows. First, we explore the impact of the two terms, patch-to-prompt and prompt-to-path, in proposed CT regularization. We find that neither of these two terms can be omitted and we attempt to choose different coefficients as discussed in Sec. C.3. Then, on Base-to-New generalization, the trade-off between performance on base and new classes is ineviTable Thus we ablate the number of training epochs on various datasets. We find that our method is very tolerant to changes in the harmonic mean and more details can be found in Sec. C. 13. Empirically, we validate that the stochastic generated module is the crucial factor affected the performance of our proposed method rather than additional parameters in inference network. We also compared the results under the OT framework to demonstrate the effectiveness of our ap-

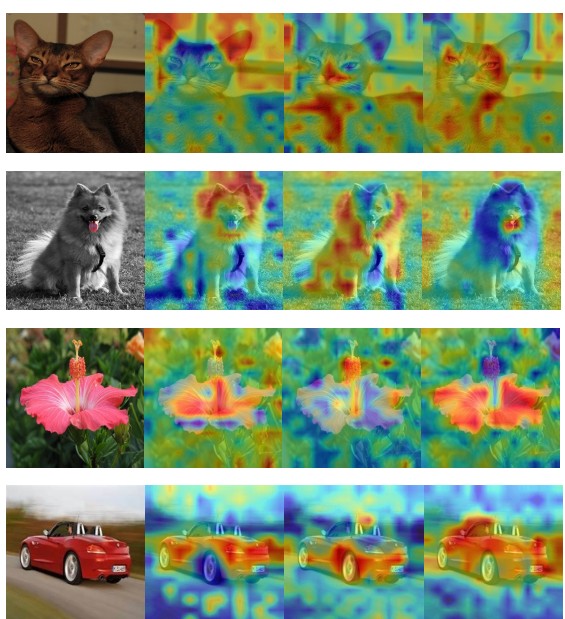

Figure 7: Visualization of the learned prompts.

proach as shown in Sec. C.10. Besides, we also evaluate the computation cost compared with other baseline methods in sec. C.11.

**Visualization** Excitingly, we have discovered that transport plans $\pi$ in Eq. 7 serve as a potent tool for achieving visualization, allowing us to demonstrate how stochastic-generated prompts for a specific class concentrate on the visual concepts of the corresponding images. We provide visualization examples in Fig 7 to illustrate this. Besides, as shown in Fig D2, we also attempt to explain the learned prompt from text domain via a multimodal model. More analysis and visualization can be found at Sec. D.

## 5 CONCLUSION

In this paper, we propose Patch-Prompts aligned Bayesian prompt tuning (PBPrompt) for pre-trained vision-language models. PBPrompt is a Bayesian prompt tuning method that generates label-specific stochastic prompts hierarchically under the variational inference framework comprising a

stochastic sampling network and a deterministic generative model. Moreover, we also introduce a CT regularization that aligns the textual prompts with the image patches under the conditional transport framework. PBPrompt is optimized by the derived combined ELBO via the stochastic gradient algorithm. Extensive experiments over 15 datasets at various tasks are conducted to evaluate the efficiency of our models. We hope PBPrompt will provide a simple tool for prompt tuning and inspire future work.

## Acknowledgements

This work was supported in part by the National Natural Science Foundation of China under Grant U21B2006; in part by Shaanxi Youth Innovation Team Project; in part by the Fundamental Research Funds for the Central Universities QTZX24003 and QTZX22160; in part by the 111 Project under Grant B18039;

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

# A    DISCUSSIONS

**The main purpose of the introduced Bayesian prompt generation and Patch-Prompt CT alignments.**

One of the main contributions of the proposed model is the stochastic prompt generation, which introduces uncertainty into the prompt embeddings. E.g., for each category, we can generate different prompts that capture diverse visual concepts, resulting in better class-specific representations. Unfortunately, due to the mode-collapse problem that usually appears in most Bayesian generative models, we find that only optimizing the stochastic module by the classification loss could lead to suboptimal results. Motivated by previous PLOT [Chen et al., 2022], we here employ the CT regularization to align the generated prompts and the image patches. Intuitively, we view images are two discrete distributions over the prompt and patch embeddings. They share similar semantics but with different domains. Ideally, those two distributions should have close semantic distance. By minimizing the CT distance, the learned prompt embeddings tend to capture the true label-specific visual concepts, improving the quality of the learned prompts. That is, the CT regularization improves the performance of the method by aligning the textual prompt domain and the visual patch domain, which is usually ignored by previous works.

**The improvement is marginal when compared to CoCoOp in some cases.**

We highlight the superiority of the proposed model below. First, the paper provides a novel Baeysian prompt-generation strategy for the prompt-tuning community. This enables the learned prompt to capture diverse visual concepts and gives the following studies a new stochastic view rather than only focusing on deterministic paradigms. Second, consistent improvement in most cases. We here want to note that it is a nontrivial contribution that achieves consistent improvement over 4 tasks on 15 datasets. For the marginal improvement on several datasets, we note that previous models (e.g., CoCoOp) have achieved high results, and thus the improvements are slight. We find that the proposed PBPrompt usually has a significant improvement on 1/2/4 shots, which clearly highlights the performance of our method with fewer training samples(see Table C. 5 and Table C. 6 for detailed results). Besides, our method balances the seen and unseen sets well according to Table 4. E.g., PBPrompt achieves 0.9%-9.14 % improvements compared to CoCoOp in terms of H score. Third, the interpretability of the proposed model. The visualization in Fig **??**(a) shows the interpretability of the learned prompts, while CoCoOp only reports the numerical results.

**Differences between SHIP.**

Both SHIP and PBPrompt introduce the uncertainty into the prompt generation process. However, the latent variable $z$ ($r$ in PBPrompt) models different levels of uncertainty and comes from different assumption. SHIP introduces the stochastic prompts into each image, and infers a sample-dependent posterior:

$$q(\boldsymbol{z}_i) = \mathcal{N}(\mu(\boldsymbol{x}_i), \Sigma(\boldsymbol{x}_i)), \tag{10}$$

where $\boldsymbol{x}_i$ denotes the feature of $i$-th image. While PBPrompt views each category has a underlying distribution and infers a label-specific posterior:

$$q(\boldsymbol{z}_c) = \mathcal{N}(\mu(\boldsymbol{e}_c), \Sigma(\boldsymbol{e}_c)), \tag{11}$$

where $\boldsymbol{e}_c$ denote the embedding of $c$-th category.

**Prior on** $p(\boldsymbol{z})$. SHIP simply adopts the standard Gaussian as the prior of $\boldsymbol{z}$, e.g., $p(\boldsymbol{z}) = \mathcal{N}(0, \mathbf{I})$, while PBPrompt utilizes the contextual prior to capture label-specific features: $p(\boldsymbol{z}_c) = \mathcal{N}(\boldsymbol{e}_c, \mathbf{I})$. This difference enables PBPrompt to access additional label semantics, achieving better prior guidance.

**Training pipelines**. SHIP introduces an additional feature reconstruction loss to pre-train the VAE, and then finetunes the prompt via the task-specific loss. Our PBPrompt naturally interages the stochastic prompts into the CLIP framework and directly optimize the prompt via the combined ELBO.

# B    METHOD DETAILS

Given the labeled training dataset $\mathcal{D} = (\boldsymbol{x}_j, y_j)_{j=1}^{N_{tr}}$, our proposed PBPrompt aims to learn stochastic prompts for each class. Note that, all parameters in PBPrompt are optimized by minimizing the combined ELBO end-to-end. We summarize the training algorithm at Algorithm 1.

---

**Algorithm 1** Training algorithm for our proposed PBPrompt.

---

**Output**: The trained PBPrompt, which can generate the stochastic label-specific prompts for downstream tasks.

**Input**: Training set $\mathcal{D} = (\boldsymbol{x}_j, y_j)_{j=1}^{N_{tr}}$, a VLP, class names, and hyperparameter $\eta$.

**Initialize**: The prefix token embeddings, the parameters in inference network $q(\boldsymbol{r}_c|c)$ and the generative model $\phi(\boldsymbol{v}_c|\boldsymbol{r}_c)$.

**for** iter = 1,2,3,... **do**

    Sample a batch of $B$ image-label pairs and get the image feature and patch embeddings by feeding the image into the image encoder $f(\boldsymbol{x})$.

    # Learning of PBPrompt

    Generate $C$ stochastic prompts hierarchically with Eq.(2) for all classes.

    Get the label embeddings by feeding the prompts into the text encoder $g(\boldsymbol{t})$.

    Compute the CT distance between patches and the class-specific prompts with Eq.(5).

    Compute the combined ELBO $\mathcal{L}$ with Eq.(8) and update all learnable parameters by minimizing the $\mathcal{L}$ with the stochastic gradient descent algorithm.

**end for**

---

## C  EXPERIMENT DETAILS

### C.1  DATA STATISTICS

Our experiments are conducted on 15 widely-used vision datasets. *E.g.*, ImageNet Deng et al. [2009] and Caltech101 Fei-Fei et al. [2004] for generic object classification, OxfordPets Parkhi et al. [2012], StanfordCars Krause et al. [2013], Flowers102 Nilsback and Zisserman [2008], Food101 Bossard et al. [2014] and FGVCAircraft Maji et al. [2013] for fine-grained image recognition, EuroSAT Helber et al. [2019] for satellite image classification, UCF101 Soomro et al. [2012] for action classification, DTD Cimpoi et al. [2014] for texture classification, and SUN397 Xiao et al. [2010] for scene recognition. For the domain generalization task, we use ImageNet as the source domain dataset and evaluate performance on ImageNetV2 Recht et al. [2019], ImageNet-Sketch Wang et al. [2019], ImageNet-A Hendrycks et al. [2021b], and ImageNet-R Hendrycks et al. [2021a]. We summarize the data statistics at Table C. 1

Table C. 1: Statistics of the datasets.

| Dataset | Classes | Train | Val | Test |
|---------|---------|-------|-----|------|
| ImageNet | 1000 | 1.28M | N/A | 50,000 |
| Caltech101 | 100 | 4,128 | 1,649 | 2,465 |
| OxfordPets | 37 | 2,944 | 736 | 3,669 |
| StanfordCars | 196 | 6,509 | 1,635 | 8,041 |
| Flowers102 | 102 | 4,093 | 1,633 | 2,463 |
| Food101 | 101 | 50,500 | 20,200 | 30,300 |
| FDVCAircraft | 100 | 3,334 | 3,333 | 3,333 |
| SUN397 | 397 | 15,880 | 3,970 | 19,850 |
| DTD | 47 | 2,820 | 1,128 | 1,692 |
| EuroSAT | 10 | 13,500 | 5,400 | 8,100 |
| UCF101 | 101 | 7,639 | 1,808 | 3,783 |
| ImageNetV2 | 1000 | N/A | N/A | 10,000 |
| ImageNet-Sketch | 1000 | N/A | N/A | 50,889 |
| ImageNet-A | 200 | N/A | N/A | 7,500 |
| ImageNet-R | 200 | N/A | N/A | 30,000 |

### C.2  HYPERPARAMETER SETTING

We set the training hyper-parameters as well as the training pipeline to be the same as Zhou et al. Zhou et al. [2022a] in terms of definitions of few-shot tasks while using ViT-B/16 in the manuscript. For the RN50 backbone, we replace the ViT-B/16 with RN50 and set the number of shots as 4 to maintain consistency with the other works using RN50. We list those settings at Table C. 2.

Table C. 2: All results in the main paper were generated using shared hyperparameters when employing the ViT-B/16 backbone.

| Hyperparameters | Values |
|---|---|
| Batch Size | 1 |
| Input Size | $224 \times 224$ |
| Input Interpolation | "Bicubic" |
| Input Pixel Mean | $[0.48145466, 0.4578275, 0.40821073]$ |
| Input Pixel STD | $[0.26862954, 0.26130258, 0.27577711]$ |
| Transforms | ["random resized crop", "random filp", "normalize"] |
| Optimizer | SGD |
| Learning Rate | $2e$-3 |
| LR Scheduler | "cosine" |
| Warmup Epoch | 1 |
| Warmup Type | "constant" |
| Warmup LR | $1e$-5 |
| Backbone | ViT-B/16 |
| Prompt Length | 4 |
| Prompt Initialization | "" |
| Precision | "fp16" |
| Number of shots | 16 |

## C.3 IMPACT OF THE PATCH-TO-PROMPT AND PROMPT-TO-PATCH TRANSPORT

In the previous experiments, we view the patch-to-prompt and prompt-to-patch transport in Eq. 6 equally. To discuss the impact of those two terms, we rewrite Eq. 6 as:

$$\mathcal{L}_{CT}(P, Q) = \lambda \mathcal{L}_{\boldsymbol{u} \to \boldsymbol{g}} + (1 - \lambda)\mathcal{L}_{\boldsymbol{g} \to \boldsymbol{u}}, \tag{12}$$

where $\lambda$ controls the weight of the patch-to-prompt term. We report the few-shot results with various $\lambda$ at Fig C. 3. We find that 1) regardless of considering the $\mathcal{L}_{\boldsymbol{u} \to \boldsymbol{g}}$ or the $\mathcal{L}_{\boldsymbol{g} \to \boldsymbol{u}}$, the final experimental results were not satisfactory. 2) Promising results could be obtained by carefully choosing $\lambda$. Thus we set this hyperparameter as 0.5 for ease of parameter tuning.

| Dataset | | 0.0 | 0.2 | 0.4 | 0.5 | 0.6 | 0.8 | 1.0 |
|---|---|---|---|---|---|---|---|---|
| | 1-shot | 51.36 | 51.54 | 51.77 | **52.03** | 51.83 | 51.95 | 51.37 |
| | 2-shots | 54.43 | 55.67 | 56.20 | **56.34** | 55.85 | 55.20 | 55.67 |
| **DTD** | 4-shots | 58.16 | 58.75 | **59.66** | 59.63 | 59.53 | 59.42 | 58.87 |
| | 1-shot | 60.78 | 61.21 | **61.93** | 60.92 | 61.02 | 61.61 | 61.20 |
| | 2-shots | 68.12 | 68.76 | 68.34 | **68.77** | 68.05 | 67.43 | 67.98 |
| **EuroSAT** | 4-shots | 70.63 | 71.01 | 71.1 | **72.84** | 72.71 | 72.14 | 71.96 |
| | 1-shot | 93.21 | 93.90 | **93.94** | 93.92 | 93.93 | 93.32 | 93.4 |
| | 2-shots | 93.98 | 94.20 | 94.41 | 94.40 | **94.45** | 94.39 | 94.23 |
| **Caltech101** | 4-shots | 94.78 | **94.85** | 94.83 | 94.83 | 94.83 | 94.80 | 94.51 |
| | 1-shot | 66.21 | 66.54 | 67.10 | **67.30** | 66.70 | 66.98 | 66.49 |
| | 2-shots | 69.52 | 70.14 | **70.48** | 70.20 | 70.36 | 70.44 | 70.23 |
| **StanfordCars** | 4-shots | 72.94 | 73.57 | 73.42 | **73.60** | 73.61 | 73.84 | 73.60 |

Table C. 3: Ablation studies of Base-to-New generalization on Bayesian prompt tuning (B-Prompt) and Patch-Prompt CT alignment (P-Prompt).

## C.4 ADDITIONAL COMPARISON TO PRODA

We compared PBPrompt to PLOT in the manuscript, and extensive results show the superiority of the proposed Bayesian framework. Note that ProDA [Lu et al., 2022] also comes from stochastic prompt tuning. We summarize the difference

below. First, ProDA focuses on the output embeddings of prompts and employs a Gaussian distribution to model the latent representation by pre-defining K label-specific templates. However, ours is a novel Bayesian prompt generation method based on input embeddings, aiming to generate the label-specific stochastic prompts in a data-driven framework, rather than based on handcraft prompts. Second, we introduce the CT regularization to align the textual prompt domain and the visual patch domain and develop a novel combined loss to optimize the proposed model end-to-end. While the ProDA employs an EM algorithm to train the parameters. Last, the learned transport plan provides us with an interpretable tool to visualize the learned prompts, while the ProDA fails to give such an interpretable.

Empirically, we report the Base-to-New comparisons (H score) at Table C. 4. Because of the unreleased code of ProDA, we could only compare with results adopted from previous work [Derakhshani et al., 2022] under the same setting on the Base-to-New task. From Table C. 4, we find that our proposed method outperforms ProDA on 9/11 datasets and has the best result on average accuracy.

Table C. 4: H score of CoCoOp, ProDA, and PBPrompt on Base-to-New task.

| Method | Imagenet | Caltech | Pets | Cars | Flowers | Food | Aircraft | SUN | DTD | EuroSAT | UCF | Average |
|---|---|---|---|---|---|---|---|---|---|---|---|---|
| CoCoOp | 73.10 | 95.84 | 96.43 | 72.01 | 81.71 | 90.99 | 27.74 | 78.27 | 64.85 | 71.21 | 77.64 | 75.83 |
| ProDA | 72.72 | 95.68 | 96.62 | 72.91 | 80.66 | 89.43 | **35.46** | 77.79 | **66.44** | 73.88 | 78.04 | 76.65 |
| PBPrompt | **73.76** | **96.66** | **96.92** | **73.02** | **83.12** | **91.22** | 34.64 | **78.35** | 66.41 | **80.34** | **79.51** | **77.86** |

## C.5 FEW-SHOT LEARNING DETAILS

In this section, we provide the complete results on few-shot learning task using ViT-B/16 and RN50 respectively. As a result of introducing additional learnable parameters into our model, we trained for more epochs that the maximum epoch is set to 400 for 16/8 shots, 200 for 4/2 shots, and 100 for 1 shot for all datasets. Table C. 5 shows more detailed accuracy consistent with Fig 3 in the manuscript. Besides, we ablate the backbone using RN50 with CoOp Zhou et al. [2022a], PLOT Chen et al. [2022], and our PBPrompt, and report the results in Table C. 6. We find that our PBPrompt also has comparable performance with other baselines, especially on 1/2/4 shots. These results, as shown in the two tables, highlight the stable performance across different backbones, demonstrating the strong robustness of our model.

Table C. 5: The few-shot learning results of various methods on 11 datasets using **ViT-B/16**. We report the average value over three different seeds.

| Dataset | Methods | 1 shot | 2 shots | 4 shots | 8 shots | 16 shots |
|---------|---------|--------|---------|---------|---------|----------|
| ImageNet | CoOp | 68.10 | 69.25 | 69.53 | 70.40 | 71.51 |
| | CoCoOp | 68.40 | 69.13 | 69.30 | 70.45 | 71.60 |
| | PLOT | 67.40 | 68.80 | 69.90 | 70.15 | 71.37 |
| | PBPrompt | 69.55 | 69.90 | 70.50 | 71.62 | 71.86 |
| Caltech101 | CoOp | 93.13 | 92.97 | 94.50 | 94.73 | 95.50 |
| | CoCoOp | 92.27 | 93.47 | 94.27 | 94.73 | 95.21 |
| | PLOT | 87.90 | 89.53 | 91.87 | 92.90 | 93.80 |
| | PBPrompt | 93.92 | 94.40 | 94.83 | 95.13 | 95.37 |
| DTD | CoOp | 50.03 | 53.93 | 59.23 | 64.37 | 68.40 |
| | CoCoOp | 50.80 | 54.10 | 58.37 | 63.07 | 67.67 |
| | PLOT | 52.20 | 56.03 | 58.37 | 65.57 | 70.17 |
| | PBPrompt | 52.03 | 56.20 | 59.63 | 64.17 | 68.50 |
| EuroSAT | CoOp | 51.80 | 66.33 | 65.87 | 74.77 | 83.07 |
| | CoCoOp | 51.93 | 64.17 | 67.20 | 75.07 | 82.87 |
| | PLOT | 59.77 | 69.03 | 73.50 | 80.03 | 83.47 |
| | PBPrompt | 60.92 | 68.77 | 72.84 | 80.14 | 84.21 |
| FGVCAircraft | CoOp | 26.20 | 27.90 | 30.03 | 36.00 | 39.73 |
| | CoCoOp | 16.83 | 26.47 | 29.27 | 36.17 | 38.60 |
| | PLOT | 20.20 | 21.87 | 23.90 | 27.13 | 30.57 |
| | PBPrompt | 27.41 | 29.03 | 31.89 | 36.10 | 39.54 |
| Flowers102 | CoOp | 73.00 | 81.90 | 86.50 | 94.13 | 96.20 |
| | CoCoOp | 76.80 | 86.40 | 91.80 | 93.98 | 96.30 |
| | PLOT | 70.50 | 80.57 | 88.70 | 93.77 | 95.70 |
| | PBPrompt | 75.43 | 83.37 | 88.90 | 94.00 | 96.32 |
| FOOD101 | CoOp | 82.70 | 82.77 | 83.63 | 84.00 | 85.33 |
| | CoCoOp | 83.35 | 82.85 | 82.75 | 84.20 | 85.46 |
| | PLOT | 69.33 | 72.73 | 75.17 | 76.70 | 77.87 |
| | PBPrompt | 85.55 | 86.25 | 86.30 | 87.00 | 87.10 |
| OxfordPets | CoOp | 90.27 | 89.93 | 92.20 | 92.47 | 92.47 |
| | CoCoOp | 90.20 | 88.87 | 91.77 | 91.73 | 92.10 |
| | PLOT | 82.93 | 85.40 | 85.97 | 87.40 | 88.10 |
| | PBPrompt | 91.20 | 91.73 | 92.63 | 93.00 | 93.40 |
| StanfordCars | CoOp | 67.03 | 70.13 | 73.27 | 76.90 | 79.13 |
| | CoCoOp | 67.13 | 68.83 | 72.03 | 76.10 | 77.45 |
| | PLOT | 45.97 | 51.43 | 53.97 | 59.62 | 64.51 |
| | PBPrompt | 67.30 | 70.20 | 73.60 | 77.23 | 79.47 |
| SUN397 | CoOp | 67.32 | 67.67 | 70.14 | 72.37 | 74.57 |
| | CoCoOp | 65.60 | 66.13 | 69.85 | 70.35 | 73.13 |
| | PLOT | 55.17 | 59.40 | 62.73 | 65.80 | 67.00 |
| | PBPrompt | 68.10 | 69.35 | 70.21 | 72.20 | 74.15 |
| UCF101 | CoOp | 70.07 | 73.30 | 77.87 | 80.10 | 82.40 |
| | CoCoOp | 70.80 | 73.50 | 76.15 | 79.23 | 82.30 |
| | PLOT | 49.63 | 53.20 | 60.80 | 67.23 | 70.50 |
| | PBPrompt | 71.45 | 74.90 | 77.60 | 79.77 | 80.93 |
| Average | CoOp | 67.24 | 70.55 | 70.02 | 76.36 | 78.92 |
| | CoCoOp | 66.74 | 70.36 | 72.98 | 75.92 | 78.43 |
| | PLOT | 60.09 | 64.36 | 67.69 | 71.48 | 73.91 |
| | PBPrompt | 69.35 | 72.19 | 74.45 | 77.31 | 79.17 |

Table C. 6: The few-shot learning results of various methods on 11 datasets using **RN50**. We report the average value over three different seeds.

| Dataset | Methods | 1 shot | 2 shots | 4 shots | 8 shots | 16 shots |
|---|---|---|---|---|---|---|
| Caltech101 | CoOp | $87.51 \pm 1.02$ | $87.84 \pm 1.10$ | $89.52 \pm 0.80$ | $90.28 \pm 0.42$ | $91.99 \pm 0.31$ |
| | PLOT | $89.83 \pm 0.33$ | $90.67 \pm 0.21$ | $90.80 \pm 0.20$ | $91.54 \pm 0.33$ | $92.24 \pm 0.38$ |
| | PBPrompt | $90.21 \pm 0.45$ | $90.86 \pm 0.24$ | $90.92 \pm 0.10$ | $91.37 \pm 0.21$ | $92.03 \pm 0.17$ |
| DTD | CoOp | $43.62 \pm 1.96$ | $45.35 \pm 0.31$ | $53.94 \pm 1.37$ | $59.69 \pm 0.13$ | $62.51 \pm 0.25$ |
| | PLOT | $46.55 \pm 2.62$ | $51.24 \pm 1.95$ | $56.03 \pm 0.43$ | $61.70 \pm 0.35$ | $65.60 \pm 0.82$ |
| | PBPrompt | $47.21 \pm 1.22$ | $52.08 \pm 0.78$ | $56.97 \pm 0.55$ | $61.84 \pm 0.21$ | $65.58 \pm 0.33$ |
| EuroSAT | CoOp | $52.12 \pm 5.46$ | $59.00 \pm 3.48$ | $68.61 \pm 3.54$ | $77.08 \pm 2.42$ | $83.69 \pm 0.47$ |
| | PLOT | $54.05 \pm 5.95$ | $64.21 \pm 1.90$ | $72.36 \pm 2.29$ | $78.15 \pm 2.65$ | $82.23 \pm 0.91$ |
| | PBPrompt | $57.34 \pm 3.12$ | $64.67 \pm 1.21$ | $73.10 \pm 1.34$ | $78.39 \pm 1.72$ | $82.20 \pm 0.32$ |
| FGVCAircraft | CoOp | $8.59 \pm 5.79$ | $16.52 \pm 2.38$ | $20.63 \pm 2.46$ | $26.63 \pm 0.86$ | $31.43 \pm 0.96$ |
| | PLOT | $17.90 \pm 0.09$ | $18.94 \pm 0.44$ | $22.36 \pm 0.42$ | $26.17 \pm 0.29$ | $31.49 \pm 0.89$ |
| | PBPrompt | $17.49 \pm 1.24$ | $18.72 \pm 0.45$ | $22.55 \pm 0.44$ | $26.71 \pm 0.31$ | $31.44 \pm 0.64$ |
| Flowers102 | CoOp | $67.98 \pm 1.98$ | $77.58 \pm 1.46$ | $86.10 \pm 1.05$ | $91.27 \pm 0.83$ | $94.49 \pm 0.40$ |
| | PLOT | $71.72 \pm 0.97$ | $81.19 \pm 0.79$ | $87.82 \pm 0.20$ | $92.43 \pm 0.25$ | $94.76 \pm 0.34$ |
| | PBPrompt | $70.84 \pm 1.23$ | $81.35 \pm 0.87$ | $87.57 \pm 0.34$ | $92.44 \pm 0.31$ | $94.60 \pm 0.24$ |
| FOOD101 | CoOp | $74.25 \pm 1.52$ | $72.61 \pm 1.33$ | $73.49 \pm 2.03$ | $71.58 \pm 0.79$ | $74.48 \pm 0.15$ |
| | PLOT | $77.74 \pm 0.47$ | $77.70 \pm 0.02$ | $77.21 \pm 0.43$ | $75.31 \pm 0.30$ | $77.09 \pm 0.18$ |
| | PBPrompt | $77.35 \pm 0.33$ | $77.93 \pm 0.12$ | $78.09 \pm 0.21$ | $77.79 \pm 0.20$ | $77.75 \pm 0.12$ |
| ImageNet | CoOp | $56.99 \pm 1.03$ | $56.40 \pm 0.87$ | $58.48 \pm 0.47$ | $60.39 \pm 0.57$ | $61.91 \pm 0.17$ |
| | PLOT | $59.54 \pm 0.16$ | $60.64 \pm 0.06$ | $61.49 \pm 0.23$ | $61.92 \pm 0.09$ | $63.01 \pm 0.13$ |
| | PBPrompt | $60.54 \pm 0.12$ | $60.72 \pm 0.09$ | $61.68 \pm 0.13$ | $62.00 \pm 0.09$ | $62.95 \pm 0.11$ |
| OxfordPets | CoOp | $85.99 \pm 0.28$ | $82.22 \pm 2.15$ | $86.65 \pm 0.97$ | $85.36 \pm 1.00$ | $87.02 \pm 0.89$ |
| | PLOT | $87.49 \pm 0.16$ | $86.64 \pm 0.06$ | $88.63 \pm 0.23$ | $87.39 \pm 0.09$ | $87.21 \pm 0.13$ |
| | PBPrompt | $87.75 \pm 0.25$ | $86.32 \pm 0.75$ | $89.08 \pm 0.23$ | $88.34 \pm 0.14$ | $88.45 \pm 0.21$ |
| StanfordCars | CoOp | $55.81 \pm 1.67$ | $58.41 \pm 0.43$ | $62.74 \pm 0.16$ | $67.64 \pm 0.06$ | $73.60 \pm 0.19$ |
| | PLOT | $56.60 \pm 0.36$ | $57.52 \pm 0.71$ | $63.41 \pm 0.29$ | $67.03 \pm 0.50$ | $72.80 \pm 0.75$ |
| | PBPrompt | $57.14 \pm 0.21$ | $57.76 \pm 0.34$ | $63.53 \pm 0.20$ | $67.64 \pm 0.12$ | $73.75 \pm 0.34$ |
| SUN397 | CoOp | $60.12 \pm 0.82$ | $59.60 \pm 0.76$ | $63.24 \pm 0.63$ | $65.77 \pm 0.02$ | $68.36 \pm 0.66$ |
| | PLOT | $62.47 \pm 0.43$ | $61.71 \pm 0.65$ | $65.09 \pm 0.43$ | $67.48 \pm 0.04$ | $69.96 \pm 0.24$ |
| | PBPrompt | $62.51 \pm 0.49$ | $63.45 \pm 0.66$ | $64.77 \pm 0.51$ | $67.35 \pm 0.08$ | $69.93 \pm 0.17$ |
| UCF101 | CoOp | $62.13 \pm 1.14$ | $64.05 \pm 0.99$ | $67.79 \pm 0.71$ | $72.71 \pm 0.50$ | $76.90 \pm 0.50$ |
| | PLOT | $64.53 \pm 0.70$ | $66.83 \pm 0.43$ | $69.60 \pm 0.67$ | $74.45 \pm 0.50$ | $77.26 \pm 0.64$ |
| | PBPrompt | $64.29 \pm 0.84$ | $66.88 \pm 0.32$ | $69.95 \pm 0.55$ | $74.86 \pm 0.47$ | $77.35 \pm 0.52$ |
| Average | CoOp | $59.56 \pm 2.06$ | $61.51 \pm 1.39$ | $66.47 \pm 1.29$ | $69.85 \pm 0.69$ | $73.31 \pm 0.42$ |
| | PLOT | $62.58 \pm 1.13$ | $65.21 \pm 0.72$ | $68.62 \pm 0.52$ | $71.23 \pm 0.51$ | $73.97 \pm 0.54$ |
| | PBPrompt | $62.97 \pm 0.86$ | $65.52 \pm 0.52$ | $68.93 \pm 0.42$ | $71.70 \pm 0.35$ | $74.18 \pm 0.29$ |

Besides, for a fair comparison, we re-run ProGrad Zhu et al. [2022] with ViT-B/16 and set the prompt length as 4 on 1/2/4 shot as shown at Table C. 7. Compared to ProGrad which only optimizes the prompt whose gradient is aligned to the CLIP knowledge, our approach aims to squeeze CLIP knowledge by finding the stochastic prompts for each class, showing greater potential in capturing diverse visual attributes and improving generalizability.

## C.6   BASE-TO-NEW GENERALIZATION DETAILS

In this section, we report the complete results on base-to-new generalization using ViT-B/16 and RN50 respectively. Table C. 8 shows more detailed accuracy consistent with Fig 4 in the manuscript. Besides, we also provide comprehensive results

Table C. 7: Comparison with ProGrad on the few-shot learning using **ViT-B/16**. We report the average value over three different seeds.

| Dataset | Methods | 1 shot | 2 shots | 4 shots |
|---|---|---|---|---|
| Caltech101 | CoOp | 93.13 | 92.97 | 94.50 |
| | ProGrad | 93.67 | 94.33 | 94.60 |
| | PBPrompt | **93.92** | **94.40** | **94.83** |
| DTD | CoOp | 50.03 | 53.94 | 59.23 |
| | ProGrad | 51.12 | 52.30 | 56.00 |
| | PBPrompt | **52.03** | **56.20** | **59.63** |
| EuroSAT | CoOp | 51.80 | 66.33 | 65.87 |
| | ProGrad | 56.65 | 60.65 | 68.70 |
| | PBPrompt | **60.92** | **68.77** | **72.84** |
| FOOD101 | CoOp | 82.70 | 82.77 | 86.50 |
| | ProGrad | 85.55 | 85.75 | 86.17 |
| | PBPrompt | **85.55** | **86.25** | **86.30** |
| SUN397 | CoOp | 67.32 | 67.67 | 70.14 |
| | ProGrad | 67.92 | 68.95 | 70.17 |
| | PBPrompt | **68.10** | **69.35** | **70.21** |
| UCF101 | CoOp | 70.07 | 73.30 | 77.87 |
| | ProGrad | **72.65** | 73.60 | 77.40 |
| | PBPrompt | 71.45 | **74.90** | **77.60** |

using RN50 with CoOp Zhou et al. [2022b], CoPLOT Chen et al. [2022], and our PBPrompt (shown in Table C. 9).

Table C. 9: The base-to-new generalization accuracy results of various baselines on 11 datasets using **RN50**. We report the average value over three different seeds, and the results are performed on a 16-shot base set and then evaluated on the held-out new class. The best results are **highlighted**. H: the harmonic mean.

| | Average | | | ImageNet | | | Caltech 101 | | | Oxford Pets | | |
|---|---|---|---|---|---|---|---|---|---|---|---|---|
| | Base | New | H | Base | New | H | Base | New | H | Base | New | H |
| CoCoOp | 75.7 | 64.6 | 69.71 | **68.3** | 63.1 | 65.60 | 95.0 | 90.0 | 92.43 | 92.3 | 94.6 | 92.44 |
| CoPLOT | **75.9** | 67.6 | 71.51 | 68.2 | 63.1 | 65.55 | **95.4** | 90.9 | 93.09 | 92.1 | 95.9 | 93.96 |
| PBPrompt | 75.3 | **69.4** | **72.23** | 68.2 | **63.3** | 65.66 | 94.5 | **92.3** | 93.39 | 92.4 | 95.9 | **94.12** |

| | Stanford Cars | | | Flowers 102 | | | Food 101 | | | FGVC Aircraft | | |
|---|---|---|---|---|---|---|---|---|---|---|---|---|
| | Base | New | H | Base | New | H | Base | New | H | Base | New | H |
| CoCoOp | 61.8 | 65.3 | 63.50 | **91.2** | 67.5 | 77.58 | 85.0 | 86.0 | 85.50 | 25.5 | 25.7 | 25.60 |
| CoPLOT | 63.2 | **66.5** | 64.80 | 89.6 | 69.2 | 78.09 | **85.0** | 85.2 | 85.10 | **25.6** | 26.6 | **26.09** |
| PBPrompt | **64.6** | 65.5 | **65.05** | 89.8 | **71.0** | **79.30** | 84.6 | **86.5** | 85.54 | 23.2 | **27.8** | 25.29 |

| | SUN 397 | | | DTD | | | EuroSAT | | | UCF 101 | | |
|---|---|---|---|---|---|---|---|---|---|---|---|---|
| | Base | New | H | Base | New | H | Base | New | H | Base | New | H |
| CoCoOp | 75.1 | 73.6 | 74.34 | **73.1** | 50.0 | 59.38 | 88.9 | 33.5 | 48.66 | 76.5 | 61.6 | 68.25 |
| CoPLOT | **75.2** | 73.2 | 74.17 | 72.6 | 51.4 | 60.19 | **91.0** | 55.3 | 68.79 | **77.4** | 66.2 | **71.36** |
| PBPrompt | 75.1 | **73.7** | **74.40** | 70.3 | **56.2** | 62.46 | 89.7 | **66.2** | 76.18 | 76.1 | **67.1** | 71.32 |

## C.7 DOMAIN GENERALIZATION DETAILS

In this section, we report the results of comparison between our method PBPrompt and PLOT on domain generalization using RN50. As shown in Table C. 11, our method has significant improvement on 3 out of 4 datasets using RN50 backbone. Besides, we add the comparison between our proposed method and VPT, SHIP on the domain generalization using **ViT-B/16**.

## C.8 CROSS-DATASET TRANSFER LEARNING DETAILS

In this section, we report the results of comparison between our method PBPrompt and other CoOp-based methods on cross-dataset transfer learning using ViT-B/16. As shown in Table C. 12, compared with these CoOp-based methods, the proposed method has significant improvement on 7 out of 11 datasets and only shows a slight drop on the others.

Table C. 8: The base-to-new generalization accuracy results of various baselines on 11 datasets using **ViT-B/16**. We report the average value over three different seeds, and the results are performed on a 16-shot base set and then evaluated on the held-out new class. The best and the runner-up results are **highlighted** and underlined. H: the harmonic mean.

| | Average | | | ImageNet | | | Caltech 101 | | | Oxford Pets | | |
| --- | --- | --- | --- | --- | --- | --- | --- | --- | --- | --- | --- | --- |
| | Base | New | H | Base | New | H | Base | New | H | Base | New | H |
| CLIP | 69.34 | 74.22 | 71.69 | 72.34 | 68.14 | 70.18 | 96.84 | 94.00 | 95.39 | 91.17 | 97.26 | 94.11 |
| CoOp | **82.66** | 63.22 | 71.65 | 76.14 | 67.88 | 71.77 | **98.00** | 89.81 | 93.72 | 93.67 | 95.29 | 94.47 |
| CoCoOp | 80.47 | 71.69 | 75.83 | 75.98 | 70.43 | 73.10 | 97.96 | 93.81 | 95.84 | 95.20 | 97.69 | 96.43 |
| CoPLOT | 77.20 | 60.38 | 67.76 | 75.97 | 69.23 | 72.44 | 96.53 | 82.86 | 89.17 | 93.45 | 79.76 | 86.06 |
| CoOp+VPT | 71.98 | 74.76 | 73.34 | 74.73 | 70.60 | 72.60 | 95.47 | 93.80 | 94.62 | 90.77 | 97.83 | 96.61 |
| CoOp+SHIP | 80.03 | 73.69 | 76.73 | 75.87 | 69.95 | 72.79 | 97.55 | 95.20 | 96.36 | 92.19 | 93.85 | 93.01 |
| PBPrompt | 81.36 | **74.65** | **77.86** | **76.90** | **70.87** | **73.76** | 97.98 | **95.37** | **96.66** | 95.83 | **98.03** | **96.92** |

| | Stanford Cars | | | Flowers 102 | | | Food 101 | | | FGVC Aircraft | | |
| --- | --- | --- | --- | --- | --- | --- | --- | --- | --- | --- | --- | --- |
| | Base | New | H | Base | New | H | Base | New | H | Base | New | H |
| CLIP | 63.37 | **74.89** | 68.65 | 72.08 | **77.80** | 74.83 | 90.10 | 91.22 | 90.66 | 27.19 | **36.29** | 31.09 |
| CoOp | **78.12** | 60.40 | 68.13 | **97.60** | 59.67 | 74.06 | 88.33 | 82.26 | 85.19 | **40.44** | 22.30 | 28.75 |
| CoCoOp | 70.49 | 73.59 | 72.01 | 94.87 | 71.75 | 81.71 | 90.70 | **91.29** | 90.99 | 33.41 | 23.71 | 27.74 |
| CoPLOT | 61.41 | 42.69 | 50.37 | 95.26 | 56.03 | 70.56 | 88.45 | 85.28 | 86.84 | 29.63 | 16.17 | 20.92 |
| CoOp+VPT | 65.27 | 75.97 | 70.21 | 72.97 | 75.90 | 74.40 | 90.37 | 91.67 | 91.01 | 29.57 | 33.80 | 31.54 |
| CoOp+SHIP | 68.57 | 73.90 | 71.14 | 94.02 | 74.40 | 83.06 | 90.54 | 91.03 | 90.87 | 34.27 | 32.33 | 33.28 |
| PBPrompt | 72.93 | 73.12 | **73.02** | 95.47 | 73.60 | **83.12** | 90.87 | 91.57 | **91.22** | 35.47 | 33.84 | **34.64** |

| | SUN 397 | | | DTD | | | EuroSAT | | | UCF 101 | | |
| --- | --- | --- | --- | --- | --- | --- | --- | --- | --- | --- | --- | --- |
| | Base | New | H | Base | New | H | Base | New | H | Base | New | H |
| CLIP | 69.36 | 75.35 | 72.23 | 53.24 | **59.90** | 56.37 | 56.48 | 64.05 | 60.02 | 70.53 | **77.50** | 73.85 |
| CoOp | 80.60 | 65.89 | 72.51 | **79.44** | 41.18 | 54.24 | **92.19** | 54.74 | 68.69 | **84.69** | 56.05 | 67.45 |
| CoCoOp | 79.74 | 76.86 | 78.27 | 77.01 | 56.00 | 64.85 | 87.49 | 60.04 | 71.21 | 82.33 | 73.45 | 77.64 |
| CoPLOT | 78.56 | 72.34 | 75.32 | 69.87 | 53.63 | 60.68 | 87.39 | 64.63 | 74.30 | 72.71 | 41.51 | 52.84 |
| CoOp+VPT | 73.77 | 77.90 | 75.77 | 57.67 | 58.70 | 58.18 | 67.97 | 71.63 | 69.75 | 73.23 | 74.63 | 73.92 |
| CoOp+SHIP | 79.54 | 75.27 | 77.35 | 74.88 | 56.88 | 64.65 | 88.63 | 66.87 | 76.22 | 81.08 | 76.85 | 78.91 |
| PBPrompt | 79.30 | **77.43** | **78.35** | 78.03 | 57.81 | **66.41** | 89.53 | 72.87 | **80.34** | 82.66 | 76.59 | **79.51** |

Table C. 10: Cross-domain generalization accuracy results of various baselines using **RN50**. $\Delta$: The improvements of the proposed model compared to PLOT.

| | | Source | Target | | | |
| --- | --- | --- | --- | --- | --- | --- |
| Method | Learnable | ImageNet | ImageNetV2 | ImageNet-Sketch | ImageNet-A | ImageNet-R |
| CoOp | ✓ | 61.91 | 54.26 | 32.47 | 21.78 | 54.21 |
| PLOT | ✓ | **63.01** | **55.11** | 33.00 | 21.86 | 55.61 |
| PBPrompt | ✓ | 62.95 | 54.77 | **34.10** | **24.85** | **59.89** |
| $\Delta$ | - | **−0.06** | **−0.34** | **+1.10** | **+2.99** | **+4.28** |

## C.9   TRADE-OFF ON BASE-TO-NEW GENERALIZATION

The number of training epochs causes the trade-off between performance on base and on new classes. Specifically, more training epochs lead better accuracy on base classes and lower it on new classes. Therefore, we training ImageNet, Caltech101, DTD, EuroSAT and Flowers102 for 50 more epochs on base-to-new task. As shown in Table C. 13, increasing the number of epochs in the training process can enhance performance on base classes while causing a slight decline on new classes. However, the changes in the harmonic mean are only marginally affected. For example, with more training epochs on Flowers102, our proposed method raises the performance on base classes by $+1.21$ and lower it on new classes by $-2.44$. This change slightly affects the harmonic mean, reducing it by $1.37\%$ which is still $0.33\%$ better than CoCoOp.

## C.10   MORE ABLATION STUDY DETAILS

In this section, we validate that the stochastic generated module is the crucial factor affected the performance of our proposed method instead of additional parameters in inference network. Empirically, we also compare the results with our purposed method under Optimal Transport (OT) framework to test the efficiency of the adopted CT module. We build two models

Table C. 11: Cross-domain generalization accuracy results of various baselines using **Vit-B/16**.$\Delta$: The improvements of the proposed model compared to PLOT.

| | | Source | Target | | | |
|---|---|---|---|---|---|---|
| Method | Learnable | ImageNet | ImageNetV2 | ImageNet-Sketch | ImageNet-A | ImageNet-R |
| CoOp | ✓ | 71.51 | 64.20 | 47.99 | 49.71 | 75.21 |
| CoOp + VPT | ✓ | 69.73 | 63.17 | 48.87 | 50.95 | 76.24 |
| CoOp + SHIP | ✓ | 70.12 | 63.23 | 48.65 | 50.77 | 77.40 |
| CoCoOp | ✓ | 71.02 | 64.07 | 48.75 | 50.63 | 76.18 |
| CoCoOp + VPT | ✓ | 70.70 | 64.23 | 49.20 | 51.33 | **77.00** |
| CoCoOp + SHIP | ✓ | 70.81 | 64.34 | 49.25 | 51.28 | 76.50 |
| PBPrompt | ✓ | **71.71** | **64.53** | **49.32** | **51.64** | 76.71 |
| $\Delta$ | - | +0.90 | +0.19 | +0.07 | +0.36 | +0.21 |

Table C. 12: Cross-dataset transfer learning accuracy results of CoOp-based method on source and target datasets using ViT-B/16. $\Delta$: The improvements of the proposed model compared to SHIP.

| | Source | Target | | | | | | | | | | |
|---|---|---|---|---|---|---|---|---|---|---|---|---|
| Method | Imagenet | Caltech | Pets | Cars | Flowers | Food | Aircraft | SUN | DTD | EuroSAT | UCF | Average |
| ProGrad | 71.50 | 94.43 | 90.14 | 65.32 | 71.88 | 86.06 | 22.94 | 67.36 | 45.73 | 45.37 | 68.21 | 65.74 |
| CoOp + VPT | 69.73 | 93.67 | 89.27 | 65.50 | 70.20 | 86.27 | 22.13 | 66.57 | **46.93** | 47.43 | 67.21 | 65.51 |
| CoOp + SHIP | - | 94.04 | 90.38 | 65.55 | 69.67 | **86.40** | 21.90 | 66.26 | 45.69 | **48.17** | 68.52 | 65.69 |
| PBPrompt | **71.71** | **94.87** | **90.62** | **66.00** | **72.44** | 86.34 | **24.82** | **67.69** | 45.62 | 47.13 | **68.83** | **66.40** |
| $\Delta$ | - | +0.83 | +0.24 | +0.45 | +2.77 | −0.06 | +2.92 | +1.43 | −0.07 | −1.04 | +0.31 | +0.71 |

Table C. 13: Base-to-new generalization accuracy results of our purposed method PBPrompt with more 50 training epochs on ImageNet, Caltech101, DTD, EuroSAT and Flowers102 using ViT-B/16. (·) denoted the difference from the original results in Table C. 8. $\Delta$: The improvements of harmonic mean compared to CoCoOp (without additional training epochs).

| | ImageNet | Caltech101 | Flowers102 | DTD | EuroSAT |
|---|---|---|---|---|---|
| Base | 76.97 (+0.07) | 98.01 (+0.03) | 96.68 (+1.21) | 80.44 (+2.41) | 91.86 (+2,32) |
| New | 70.12 (-0.75) | 94.43 (-0.94) | 71.16 (-2.44) | 52.15 (-5.66) | 68.08 (-4.79) |
| H | 73.36 (-0.40) | 96.19 (-0.47) | 81.98 (-1.14) | 63.28 (-1.57) | 78.20 (-2.14) |
| $\Delta$ | +0.26 | +0.35 | +0.27 | −1.57 | +6.99 |

denoted by PBPrompt$_{\text{w/o-S}}$ and PBPrompt$_{\text{OT}}$ respectively for comparison. PBPrompt$_{\text{w/o-S}}$ denotes the model removing the stochastic prompt generation process and only preserving the inference network. PBPrompt$_{\text{OT}}$ denotes the model replace the CT framework with OT framework. Then, we conduct the ablation study on the few-shot task (1/2/4 shots) with ImageNet, Caltech101, Flowers102, DTD and EuroSAT.

## C.11 COMPUTATION COST EVALUATION

In this section, we summarize the comparison of the parameters and inference speed of the baseline methods CoOp Zhou et al. [2022b], CoCoOp Zhou et al. [2022a], PLOT Chen et al. [2022] with 4 prompts and our PBPrompt with 10 samples. We report the number of learnable parameters and the number of images processed by the model in 1 second during inference on the Food101 Bossard et al. [2014] dataset. As shown in Table C. 15, despite the introduction of additional learnable parameters in our model, we were able to achieve comparable inference speed.

Table C. 14: The results of ablation study on five datasets using ViT-B/16. We report the average value over three different seeds. The best results are highlighted.

| Dataset | Methods | 1 shot | 2 shots | 4 shots |
|---|---|---|---|---|
| ImageNet | CoOp | 68.10 | 69.25 | 69.53 |
| | PBPrompt$_{w/o-S}$ | 68.27 | 69.30 | 69.92 |
| | PBPrompt$_{OT}$ | 69.03 | 69.79 | 70.23 |
| | PBPrompt | **69.55** | **69.90** | **70.50** |
| Caltech101 | CoOp | 93.13 | 92.97 | 94.50 |
| | PBPrompt$_{w/o-S}$ | 92.86 | 93.91 | 94.51 |
| | PBPrompt$_{OT}$ | 93.39 | 93.76 | 94.62 |
| | PBPrompt | **93.92** | **94.40** | **94.83** |
| Flowers102 | CoOp | 73.00 | 81.90 | 86.56 |
| | PBPrompt$_{w/o-S}$ | 73.56 | 82.04 | 87.00 |
| | PBPrompt$_{OT}$ | 74.16 | 82.66 | 87.92 |
| | PBPrompt | **75.43** | **83.37** | **88.90** |
| DTD | CoOp | 50.03 | 53.93 | 59.23 |
| | PBPrompt$_{w/o-S}$ | 50.65 | 54.55 | 59.40 |
| | PBPrompt$_{OT}$ | 51.95 | 55.66 | 59.50 |
| | PBPrompt | **52.03** | **56.20** | **59.63** |
| EuroSAT | CoOp | 51.80 | 66.33 | 65,87 |
| | PBPrompt$_{w/o-S}$ | 52.15 | 66.97 | 68.19 |
| | PBPrompt$_{OT}$ | **61.10** | 67.21 | 71.77 |
| | PBPrompt | 60.92 | **68.77** | **72.84** |

Table C. 15: The parameters and inference time comparison.

| Settings | CoOp | CoCoOp | PLOT(N=4) | PBPrompt |
|---|---|---|---|---|
| # Params | 2048 | 35360 | 8192 | 1577984 |
| Inference Speed(images/s) | 645 | 37 | 583 | 541 |

# D VISUALIZATION DETAILS

## D.1 ANALYSIS FOR VISUALIZATION

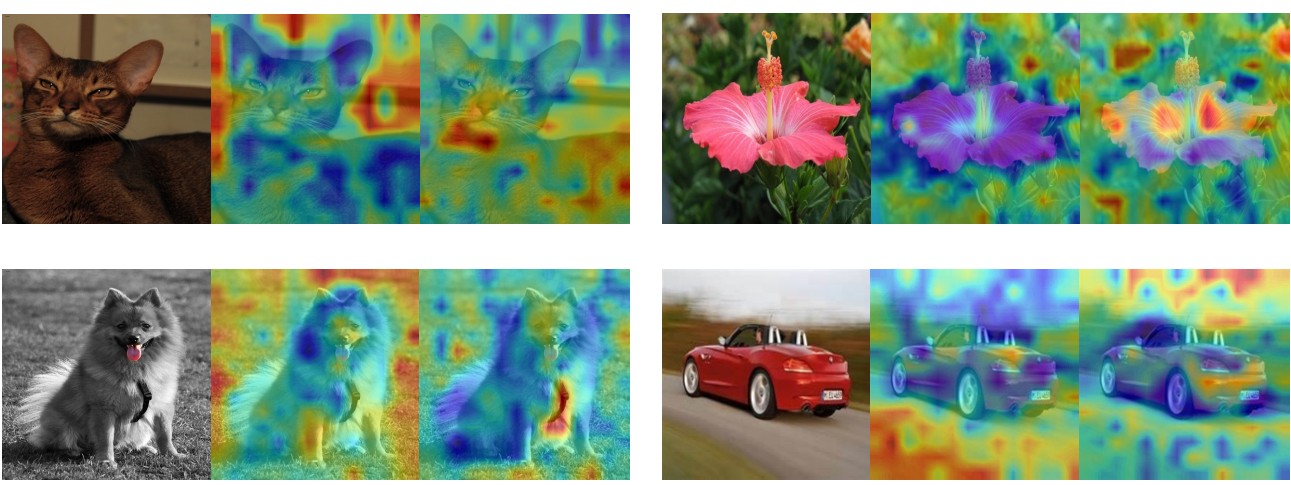

Figure D1: Visualization of the learned prompts unrelated to the corresponding class.

| | | Prompt #1 | Prompt #2 | Prompt #3 |
|---|---|---|---|---|
| | Top-1 | The crocodile stands on the edge of a body of water, which has links to the landmass that it is standing on. | Its skin is rough and scaly, with a dark brown color. | The alligator has its mouth open and its teeth visible. |
| | Top-2 | The background is a body of water. | The crocodile's body is long and slender, with a broad, flat tail. | The crocodile's body is long and slender, with a broad, flat tail. |

| | | Prompt #1 | Prompt #2 | Prompt #3 |
|---|---|---|---|---|
| | Top-1 | The elephant's body is large and muscular, with a thick trunk and large ears. | Its skin is rough and The elephant has large tusks and appears to be looking at something in the distance. | The trees in the background are tall and leafy, with branches reaching up towards the sky. |
| | Top-2 | The elephant is standing on its hind legs, with its front legs on the ground. | The elephant's body is large and muscular, with a thick trunk and large ears. | The elephant's skin appears to be brown and rough, with a few patches of dirt on its body. |

Figure D2: Prompt-caption retrieval results.

To exhibit how stochastic-generated prompts for a certain class focus on the visual concepts of the images related to the corresponding class, we have provided some visualization examples at Fig 7 in the manuscript via employing the transport plans $\pi$ to match the relations between various textual prompts and visual patches. In the first two rows, we present two images belonging to the "Abyssinian" and "Keeshond" respectively in OxfordPets. Obviously, from the heatmaps, the prompts generated from the corresponding class prefer to focus on their ears, nose, eyes, and other body parts with category-specific characteristics. In the third row, we select an image belonging to the "Hibiscus" in OxfordFlowers and the stochastic-generated prompts pay more attention to its stems, stamens, and petals. Simultaneously, we take an image belonging to the "Bentley Continental Supersports Conv. Convertible 2012" in StanfordCars in the fourth row, and the corresponding prompts concentrate on the car's body, wheels, and roof.

For the prompts generated for classes unrelated to the image, we also provided some examples to demonstrate the content they focused on. As shown in Fig D1, most heatmaps concentrate on the environment of the object, while others pay attention to certain areas of the object but lack a significant correlation with the object category attributes.

To explain the learned prompt from the text domain, one of the direct ways is to visualize the most semantically close words of the generated prompts. Unfortunately, previous works find that the most of retrieved words failed to explain the prompts [Zhou et al., 2022b]. To this end, we here adopt Mini-GPT4 to generate diverse captions and report the top-2 captions of each learned prompt according to their cosine similarity (calculated by their CLIP features) at Fig D2. From the results, we find that 1) The learned prompts indeed capture diverse label-specific concepts; 2) The retrieved captions of each prompt share close semantics, which demonstrates the coherence of the learned prompts.