# OpenReview forum: "Patch-Prompt Aligned Bayesian Prompt Tuning for Vision-Language Models"
_auai.org/UAI/2024/Conference — UAI 2024 poster_

### Official Review · Reviewer_mybj · 2024-03-04

**Q2-1 Originality-Novelty:** 2
**Q2-2 Correctness-Technical Quality:** 3
**Q2-5 Clarity Of Writing:** 3

**Q1 Summary And Contributions:**

The paper proposed a bayesian prompt tuning method, which focuses on the label-specific stochastic prompts. A hierarchical prompt generator is developed and optimized by ELBO. The conditional transport (CT) regularization is used to avoid over-fitting. Several experiments, cross four task and 15 datasets, show promising transferability and generalization performance.

**Q2-3 Extent To Which Claims Are Supported By Evidence:**

2: Fair: the main claims are somewhat supported by evidence (but the experimental evaluation may be weak, or does not match entirely with the claims, important baselines may be missing, proofs contain important ideas but lack rigor, algorithmic details are only discussed superficially, references are imprecise, assumptions are not sufficiently motivated or explicated, etc.).

**Q2-4 Reproducibility:**

3: Good: key resources (e.g. proofs, code, data) are available and key details (e.g. proofs, experimental setup) are sufficiently well-described for competent researchers to confidently reproduce the main results.

**Q3 Main Strengths:**

1. a new label-specific stochastic prompts for distributed prompt tuning.
2. improved transferability and generalization.
3. extensive experimental evaluation

**Q4 Main Weakness:**

1. The motivation for “label-specific” are not highlight.
2. No enough comparison with recent works.

**Q5 Detailed Comments To The Authors:**

1. The author should emphasize the advantages of “label-specific” compared to ”instance-specific”[1]. Figure 1 in the paper seems to be an illustration of instance-specific as one image corresponds to one prompt. In fact, I have a question whether it is possible to generate a prompt for an image that is related to its category but has another characteristic description? For example, in Figure 1 of the paper, the lower right prompt is generated for the upper right image.
2. The author cites four previous most related works (SHIP[1], ProDA[2], VPT[3], PLOT[4]) in Section 3. But there was not enough comparison in the experiment, except for the base-to-new generalization task.
** Minor: The Contextual Prior p(t^c) in Section 2.3 is not used in other parts. Maybe there is a symbol error.

[1]. Wang Z, Liang J, He R, et al. Improving zero-shot generalization for clip with synthesized prompts[C]//Proceedings of the IEEE/CVF International Conference on Computer Vision. 2023: 3032-3042.
[2]. Lu Y, Liu J, Zhang Y, et al. Prompt distribution learning[C]//Proceedings of the IEEE/CVF Conference on Computer Vision and Pattern Recognition. 2022: 5206-5215.
[3]. Derakhshani M M, Sanchez E, Bulat A, et al. Variational prompt tuning improves the generalization of vision-language models[J]. arXiv preprint arXiv:2210.02390, 2022.
[4]. Chen G, Yao W, Song X, et al. Prompt learning with optimal transport for vision-language models[J]. arXiv preprint arXiv:2210.01253, 2022.

**Q9 Complying With Reviewing Instructions:**

Yes

---

> ### Author Rebuttal · Authors · 2024-04-07
>
> We thank reviewer mybj for the comments and suggestions. Below, we address the concerns raised in your review.
> > ### **The motivation for “label-specific”**
>
> One of the core ideas of prompt tuning is to find the optimal label feature, and we empirically find that there are diverse prompts to describe a class (such as the dog) from different perspectives. Further, we also find that the prompts for the dog may be quite different from those for the car. That’s why we aim to learn stochastic prompts in a label-specific manner. Compared to instance-specific methods, we first would like to note that a class may share K concepts (K<<N, where N is the number of training images), and thus K concepts are enough to represent a class. Learning instance-specific prompts may overly focus on the visual feature, overfitting to the training images. Besides, the instance-specific methods, such as [1] need to employ a VAE to reconstruct the visual feature, which seems a little bit complex during the training stage.
> The core idea in Figure 1 is to demonstrate the diverse prompts for one label, and the picture associated with each prompt aims to provide an intuitive visualization of the corresponding prompt. Technically, the generated prompt may not related to the current image (both in baselines and ours), and we have shown some failure cases in the Appendix.
>
> [1]. Wang Z, Liang J, He R, et al. Improving zero-shot generalization for clip with synthesized prompts
>
> > ### **Comparison with recent works**
>
> The most related work to ours is PLOT, which applies optimal transport theory to learn multiple local prompts and achieve SOTA on several tasks. In this paper, We compare our proposed method with PLOT (including CoPLOT, a conditional version) on all datasets for all tasks including few-shot learning, base-to-new generalization, domain generalization, across datasets transfer learning and we also verify the robustness of our proposed method under different backbones by comparing with PLOT using two different backbones in Appendix. Additionally, we also compare our proposed method with other distributed prompt tuning methods. Specifically, we first compare our proposed method with VPT, SHIP and ProDA on base-to-new generalization in Table C.7 and then with VPT, SHIP on the cross-dataset transfer learning in Table C.4 and Table C.8 in Appendix. We have a detailed discussion about the difference between our proposed method and SHIP in Section A in the Appendix. Following your suggestions, here we add the comparison between our proposed method and VPT, SHIP on the domain generalization in R1. We will add other results in the revision.
>
> **Table R1 Comparison with VPT and SHIP on domain generalization**
> |  Methods  | ImageNet | ImageNetV2| ImageNet-S  | ImageNet-A|ImageNet-R|
> |--|--|--|--|--|--|
> | CoOp | 71.51 |64.20| 47.99 |49.71| 75.21|
> | CoOp+VPT | 69.73 |63.17| 48.87| 50.95| 76.24|
> | CoOp+SHIP | 70.12 |63.23| 48.65| 50.77| 77.40|
> | CoCoOp |71.02| 64.07 |48.75| 50.63 |76.18|
> | CoCoOp+VPT |70.70| 64.23| 49.20 |51.33| **77.00**|
> | CoCoOp+SHIP |70.81| 64.34| 49.25 |51.28| 76.50|
> | PBPrompt |**71.71**| **64.53** |**49.32**|**51.64**| 76.71|
>
> We apologize that there is a symbol error of the prior defined in Section 2.3. The letter c should be a subscript and the prior p(t_c) should be p(r_c). The prior is used in calculating KL divergence in eq8.
>
> Thanks for your helpful suggestions again! Please let us know if you have any further concerns or whether this adequately addresses all the issues that you raised with the paper. We look forward to having more conversations with you about improving our paper once more.

---

### Official Review · Reviewer_RgmK · 2024-03-04

**Q2-1 Originality-Novelty:** 2
**Q2-2 Correctness-Technical Quality:** 3
**Q2-5 Clarity Of Writing:** 2

**Q1 Summary And Contributions:**

1. The author proposes using hierarchical pathways to generate random prompts for specific labels to handle different visual concepts.
2. The author introduces regularization terms to align patches and prompt embeddings, enabling the model to leverage visual knowledge for prompt learning.
3. The author formulates the model as a variational inference problem and proposes a new loss function to optimize the model.

**Q2-3 Extent To Which Claims Are Supported By Evidence:**

3: Good: the main claims are supported by convincing evidence (in the form of adequate experimental evaluation, proofs, (pseudo-)code, references, assumptions).

**Q2-4 Reproducibility:**

3: Good: key resources (e.g. proofs, code, data) are available and key details (e.g. proofs, experimental setup) are sufficiently well-described for competent researchers to confidently reproduce the main results.

**Q3 Main Strengths:**

1.	The authors' research motivation is very meaningful, and this is clearly reflected in the paper. The paper is very well written and clear.
2.	The experiment is very detailed. The authors present many comparison experiments and ablation experiments.

**Q4 Main Weakness:**

1. I believe the authors could better articulate their work, especially in the methodology section.
2. Prompt fine-tuning methods are no longer limited to manual prompt templates; there are already many generative prompt template methods and continuous prompt methods, such as P-tuning, Prefix-tuning, adaprompt, etc. I suggest the authors compare their method with these prompt-tuning methods used for text tasks to illustrate the advantages of their approach over them.

**Q5 Detailed Comments To The Authors:**

1. The authors argue that treating prompt engineering as a point estimation problem in existing work is unreasonable as it would lose the diversity of classification features. Undoubtedly, this holds true for visual prompts, hence the introduction of the SPG module by the authors. However, for textual prompts, each concept is deterministic and unique in the semantic space. With this consideration in mind, I am very curious whether this method in the paper would negatively impact the text understanding capability of VLP.
2. I find this work valuable, but I believe Figure 2 and its corresponding description could be improved. What is the relationship between $e_c$ and $r_c$ in Figure 2? The process of generating prompt $t_c$ through hierarchical paths seems to lack detailed explanation.
3. Would directly appending the embedding of the corresponding category label $e_c$ at the end of the prompt lead to cheating by the model, causing it to focus solely on $e_c$ and neglect the role of the prompt $v_c$?

**Q9 Complying With Reviewing Instructions:**

Yes

---

> ### Author Rebuttal · Authors · 2024-04-07
>
> We thank reviewer Rgmk for the comments and suggestions. Below, we address the concerns raised in your review.
>
> > ### **Q4.1 Methodology section.**
>
> Thank you for your careful reading! We will check all the formulas and simplify their mathematical annotations in the methodology section in the revision.
>
> > ### **Q4.2 Comparison with prompt-tuning methods used for text tasks.**
>
> We appreciate your constructive suggestions. It is an interesting direction to apply our stochastic prompt generation algorithm to NLU tasks, and we believe that our PBPrompt not only captures visual concepts, but also has potential in extracting diverse sentence representations. We in this paper mainly test our approach within VLP tasks, and extensive comparison with existing SoTa baselines across 4 tasks over 15 datasets demonstrate the efficiency of the proposed model. We leave the NLU applications as a future work.
>
>
> > ### **Q5.1 Whether this method in the paper would negatively impact the text understanding capability of VLP?**
>
> For text prompts, we want to note that each prompt is generated hierarchically from a class-specific underlying distribution. Once trained on training pairs, the posterior distribution has the ability to capture the uncertainty of the corresponding label, and the sampled text prompts thus extract diverse semantics. At the test stage, we average multiple sampled prompts as the final textual prompt (See Fig 5 in the manuscript), and this super prompt captures diverse semantics, resulting in robust label representation.
>
> > ### **Q5.2 Relationship between $e_c$ and $r_c$.**
>
> We first appreciate your recognition of our work and we apologize for the confusion in Figure 2. In Figure 2, $r_c$ denotes the sampled vector from a class-specific distribution, whose mean the variable is obtained by feeding $e_c$ into a forward network. $e_c$ denotes the embedding of c-th class. Following your suggestions, we will check the mathematical annotations and elaborate them in the revision.
>
> > ### **Q5.3 Directly appending the embedding of the corresponding category label $e_c$ at the end of the prompt.**
>
> Following previous works (CoOp, CoCoOp and PLOT), we append $e_c$ at the end of the prompt and focus on the learning of prefix tokens. The prefix tokens play a core role in guiding the pre-trained language model to extract useful textual features, like “A photo of a”. By learning these vectors in a data-driven manner, the textual inputs can be better adapted according to downstream tasks, and squeeze linguistic knowledge encoded in pre-trained CLIP beyond $e_c$.
>
>
> Thanks for your helpful comments again! We look forward to having more conversations with you about improving our paper once more.

---

### Official Review · Reviewer_1pvX · 2024-03-22

**Q2-1 Originality-Novelty:** 3
**Q2-2 Correctness-Technical Quality:** 4
**Q2-5 Clarity Of Writing:** 4

**Q1 Summary And Contributions:**

This paper addresses the task of image classification, especially under practically important learning and inference scenarios such as few-shot, distribution shift, base-to-new. It builds upon opportunities afforded by joint image-language embedding spaces such as in CLIP models.

It develops a novel prompt tuning method where prompts are class-specific. Classification eventually is conduced by least-distance calculation between image embedding and fine-tuned class-specific prompts. It picks up from forerunner method CoCoOp, joining it with published stochastic prompt tuning methods. It proposes to generate class-specific stochastic prompts hierarchically by sampling a vector r_c from a learnt distribution, prepending fixed w, sending the result through a single self-attention layer, and appending class name embedding e_c.

Besides hierarchical Bayesian prompt tuning, it addresses the regularization problem of resulting prompts by proposing a conditional transport loss between image patches and prompt tokens.

The paper conducts experiments over 4 scenarios demonstrating superiority, as well as complete ablations.

**Q2-3 Extent To Which Claims Are Supported By Evidence:**

3: Good: the main claims are supported by convincing evidence (in the form of adequate experimental evaluation, proofs, (pseudo-)code, references, assumptions).

**Q2-4 Reproducibility:**

4: Excellent: key resources (e.g. proofs, code, data) are available and key details (e.g. proof sketches, experimental setup) are comprehensively described for competent researchers to confidently and easily reproduce the main results.

**Q3 Main Strengths:**

The problem addressed, task classification under various scenarios involving unseen classes, is important. (I think the setting could be generalized to other image-language tasks, but the paper doesn't address this.)

The paper is very clear to read, especially sections 1-3. Code is provided (I haven't ran it).

The technical contribution is non-trivial and works through several obstacles, selecting informed, sound solutions (CT, variational network)

Experiments and ablations are very thorough and generally convincing, with quite substantial Supplementary material.

**Q4 Main Weakness:**

Experimental comparisons against competitor published methods (other than forerunners CoOp, CoCoOp) are conducted only for Base-to-new table 4, but one would definitely like to see the results for the other scenarios too. No significance results are presented, hindering a proper assessment of whether small score differences mean anything at all in results tables.

Impact can be argued to be incremental wrt forerunner methods which are already very good.

**Q5 Detailed Comments To The Authors:**

(Unfortunately, the draft doesn't contain any line numbers, which makes referencing specific sentences very hard.)

# Repeat/general minor issues
- Table. 1 -> Table 1, Algorithm. 1 -> Algorithm 1 etc
- Algorithm 1 is absent from the paper. It must be pointed out that it appears only in the Supplementary material.
- Writing and understandability are great in 1 and 2, but drop notably in sec4 which deserves accurate editing.
- Language: we set X as Y -> we set X to Y
- The introduction/context setting doesn't spell out clearly enough that this paper is about image classification and depends/rests on capabilities of CLIP-like models, in particular their characteristic joint image-language embedding space. It might be interesting to comment on whether and how this work generalizes to other tasks?

# Minor issues
In the order they appear in the paper.

- "X X X X {class}" is misleading, is suggests repeat X tokens and using { and } as tokens too.
- fig1: black and white: unfinished sentence
- Kingma and Welling reference has no date
- class can be denoted as -> can be noted
- fig 2: variantial -> variational
- fig2, it might help understanding to draw a box around v_c, e_c labelled t_c, and similarly r_c, w labelled s_c
- end, One
- Greff citation: add []
- Eq3: if we write w_1+PE_2, the last index should be PE_{b+1}. w is called prefix, but for the purpose of the PE eq3 shows it is a suffix.
- Self-Atten, Self-Attn: normalize symbol
- label token prevented: word choice prevented doesn't make sense here; prepended to the beginning??
- class-specific prefix sequence: for clarity, specify "v_1...v_b"
- before eq4: discrete distributions: specify "over the joint latent image-language embedding space"
- patch embeddings and prompt embeddings: for clarity add P and Q
- aims at best utilize
- SynHesized: missing t
- lebel
- must mention Supplementary when referencing tables which are absent from the paper
- Baselines paragraph ends with ,
- we re-run them in the same setting that: fix entire sentence
- Note that, the -> that the
- shown at Table -> in
- exhibits large gaps: do you mean performance improvement Delta? ("gaps" seems inappropriate here)
- concerns about the: rewrite sentence
- performs the best accuracy: rewrite
- useµ -> use µ
- fewer samples leads to increased uncertainty: what does this mean at all?
- introduced as follows -> introduced.
- proposed method instead of: do you mean "rather than"?
- basseline

# Questions
- is e_c defined as average token embeddings for all tokens composing the name of the class in natural language, eg for class name "fox terrier" tokens might be [fox_, te, rr, ier]?
- unnumbered eq after eq1: is p = (p(y=c|x))_c ?
- prefix token sequence: why does d_l depend on l ?
- eq4 is this p_c from eq1? so it depends on f(x)?
- eq8: replace pi by phi?
- Contextual prior: is this p(t_c) (subscript)? where does it appear?
- with 16 shots: what are shots? MC samples?
- prompt length: is this b? (rather specify in text!)
- learnable prompt embedding vectors: is this r_c (rather specify in text)
- 3.14%, more generally all result tables: how many significant digits do we have? what statistical significance do we have? you have used 3 seeds, so have an estimate of variance
- table C13: could you explain the parameter count? it seems high

**Q9 Complying With Reviewing Instructions:**

Yes

---

> ### Author Rebuttal · Authors · 2024-04-07
>
> We thank reviewer 1pvX for your positive comments and valuable suggestions, which help us improve our paper quality.  Below, we address the concerns raised in your review. Please let us know if you have any further concerns or whether this adequately addresses all the issues that you raised with the paper.
>
> ## **Rebuttal for main weakness**
> > ### **Q4.1 Missing results of competitor methods**
>
> In the main body and the supplementary material of the submitted manuscript, we have reported the comparison results of PBPrompt and baselines (CoOp, CoCoOp, PLOT, VPT, SHIP and ProDA) on 3/4 tasks: Base-to-New (Table C.7), Cross-domain (Table C.9) and Cross-datasets (Table C.10). We will reorganize the experiment so that the results in the supplementary are added in the main body, and we will also add missing results of comparison with VPT and SHIP in the revision based on your suggestions.
>
> > ### **Q4.2 No significance results are presented & Impact can be argued to be incremental**
>
> We understand the reviewer's concern about the marginal improvements. We highlight the superiority of the proposed method below.
> 1. The paper provides a novel Baeysian prompt generation strategy for the prompt tuning community. This enables the learned prompt to capture diverse visual concepts and gives the following studies a new stochastic view rather than only focusing on deterministic paradigms.
> 2. Consistent improvement in most cases. We here want to note that it is a nontrivial contribution that achieves consistent improvement over 4 tasks on 15 datasets. For the marginal improvements on several datasets, we note that previous models (e.g., CoCoOp, PLOT) have achieved high results, and thus the improvements are slight. We find that the proposed PBPrompt usually has a significant improvement on 1/2/4 shots, which clearly highlights the performance of our method with fewer training samples(see Table C. 5 & 6 in the Appendix for detailed results). Besides, our method balances the seen and unseen sets well according to Figure 4 in the manuscript.
> 3. Interpretability of the proposed model. The visualization in Figure 7 shows the interpretability of the learned prompts, while CoCoOp and CoOp only report the numerical results.
>
> >## **Minor issues**
>
> We appreciate your very careful reading of this paper and your very detailed advice!
> We will follow your advice to check for these typos and misleading presentations in the manuscript and correct them in the revision!
>
> >## **How can this work be generalized to other missions?**
>
> we first appreciate your attention to such methods. There are still many scenarios that require the ability to align the semantics between modalities, such as the text-to-image (conditional) generation task that has been very popular lately, which requires the semantic alignments between text and image, where the textual prompts guide the generation of the visual patches at each timestep. We find that our CT-based alignments may have a potential application in such tasks, and we will leave this as a future study.
>
> >## **Answers for other questions**
>
> *  $e_c$ here is defined as average token embedding of all class tokens.
> * $\mathcal{L}(p)$ →$\\mathcal{L}(v)$, $v=\\{v_{c,l}\\in\\mathbb{R}^{d_l}\\}^b_{l=1}$ → $v = \\{v_{c,l} \\in \\mathbb{R}^d\\}_{l=1}^b$
> * Yes, $p_c$ in Eq 4 is defined by Eq 1 and it depends on $f(x)$
> * There should be $\phi$ in Eq 8
> * Sorry for the typo, and the contextual prior $p(t^c)$ should be corrected as $p(r_c)$. It is a pre-defined Gaussian prior and is used to calculate the KL term in Eq 8.
> * In few-shot learning, 16 shots denote 16 samples are taken under each class while training.
> * b denotes the length of the prompt.
> * The learnable prompt embedding vectors is $v$. $r_c$ denotes the sampled vector from class-specific distribution $p(r_c)$
> * The specific results of the improvement are calculated from Table C.5 in the Appendix. We also report the detailed results with variance using RN50 in Table C.6 in the Appendix.
> *  The parameter counts are as follows:
> | Params name | number of Params |
>  |--|--|
> |Learnable prompt vector| $4\times512$|
> |inference network in SGM: linera1.weight |$1024\times512$ |
> |inference network in SGM: linera1.bias |$1024$ |
> |self-attention network in SGM: attn.in_proj_weight |$1536\times512$ |
> | self-attention network in SGM: attn.in_proj_bias|$1536$ |
> |self-attention network in SGM: attn.out_proj_weight |$512\times512$ |
> | self-attention network in SGM: attn.out_proj_bias|$512$ |
> | Total  | $1577984$   |
>
> Thanks for your helpful suggestions again! We look forward to having more conversations with you about improving our paper once more.

---

### Official Review · Reviewer_uJsA · 2024-03-24

**Q2-1 Originality-Novelty:** 2
**Q2-2 Correctness-Technical Quality:** 3
**Q2-5 Clarity Of Writing:** 3

**Q1 Summary And Contributions:**

This paper proposes to model the text prompt in CLIP-style visual-language models from a Bayesain perspective, i.e., viewing it as a distribution instead of a single optimal value. They use a hierarchical model where the first-level is stochastic. They also propose new loss functions based on optimal transport.

**Q2-3 Extent To Which Claims Are Supported By Evidence:**

2: Fair: the main claims are somewhat supported by evidence (but the experimental evaluation may be weak, or does not match entirely with the claims, important baselines may be missing, proofs contain important ideas but lack rigor, algorithmic details are only discussed superficially, references are imprecise, assumptions are not sufficiently motivated or explicated, etc.).

**Q2-4 Reproducibility:**

2: Fair: key resources (e.g. proofs, code, data) are unavailable but key details (e.g. proof sketches, experimental setup) are sufficiently well-described for an expert to confidently reproduce the main results.

**Q3 Main Strengths:**

1. The writing is easy to follow. They illustrate their motivation clearly.

2. They have careful designs for the new Bayesian framework.

**Q4 Main Weakness:**

1. Limited novelty. The method is a marginal design based on previous CLIP works.

2. Negligible improvement. It seems that the improvement in performance is also marginal besides the design itself. Is there a direct comparison between the Bayesian version and the original version?

3. Insufficient ablation studies. Among the fundamental designs is the reverse optimal transport loss term. However, the advantage of this term is not discernible from Fig 6. The smallest $\eta$ observed is 0.005, while the optimal value of $\eta$ is 0.01, showing comparable performance to 0.005. Nevertheless, as $\eta$ increases, there is a sharp decline in performance. This indicates that this term negatively impacts performance. It remains unclear what the performance would be when setting $\eta$ to 0.

**Q5 Detailed Comments To The Authors:**

See above.

**Q9 Complying With Reviewing Instructions:**

Yes

---

> ### Author Rebuttal · Authors · 2024-04-07
>
> We would like to thank our reviewer KBov for your valuable comments and suggestions. Please check our responses to your concerns point by point. We hope the following replies will change your initial assessment.
>
> > ### **Q4.1 Novelty of this paper.**
>
> First, we want to point out that efficiently tuning the pre-trained vision-language model, such as CLIP, to downstream tasks remains a challenging problem in the community. How to find the optimal and robust prompts that capture the representative features of the corresponding class is the key to the prompt tuning-based algorithm.
> We in this paper give a novel solution under the Bayesian framework and aim to explore class-specific stochastic prompt tuning. The resulting PBPrompt is different from previous works in terms of prompt generation and technical tricks to avoid overfitting.
>
> **Stochastic Prompts Generation**. Unlike deterministic prompt tuning methods which model each class as a point feature, failing to capture diverse visual attributes, we first introduce the concept of stochastic prompts generation (SPG) to view each class as a distribution at the prompt embedding space. A prompt is generated by first sampling a seed vector from the class-specific underlying distribution and then applying a lightweight language model. SPG integrates the uncertainty into prompt tuning and enables PBPrompt to capture diverse visual attributes, resulting in robust label representation. Table 3, 4, and Fig 7, D2 in the manuscript demonstrate the impact of the SPG module and diverse prompt visualization, respectively.
>
> **Alignment between textual prompts and visual patches**. Overfitting to the training patterns is another issue in prompt tuning, where the prompts tend to learn shortcuts in seen class names and fail to generalize to the unseen class. We in this paper develop a novel semantic regularization approach based on conditional transport (CT) framework. CT first views textual prompts and visual patches as two discrete distributions and minimizes their transport distance to align the semantics across modalities. This CT regularization guides the learned prompts to capture the true visual concepts (e.g., transport to the related patches) and thus mitigate the overfitting issue.
>
> > ### **Q4.2 Improvement over the original version.**
>
> We understand the reviewer’s concern about the marginal improvements, and we would like to note that
>
> 1) The current outstanding works have already achieved high performance in some settings, yet it is a nontrivial contribution to achieve consistent improvement across four tasks over 15 datasets.
>
> 2) Expect for the cross-dataset and cross-domain generalization tasks, where models are trained on large ImageNet dataset, our proposed PBPrompt also achieves visible improvements on  two other tasks, e.g., PBPrompt achieves average 2.11, 1.46, 1.47, 1.39 and 0.25 improvements over the second methods on 1,2,4,8,16 shots learning tasks; and achieves an average 2.03 improvement in terms of H score on base-to-new tasks.
>
> To test the impact of the introduced Bayesian module, we compared PBPrompt with its variant  P-Prompt (P-Prompt removes the Bayesian generation module) on few-shot learning and base-to-new tasks at Table 3 and Table 4 in the manuscript.
> As shown in Table 3 and Table 4, we find that PBPrompt achieves average 0.72, 1.31 and 1.35 improvements on 1,2,4 shots learning; and achieves an average 3.96 improvement in terms of H score on the base-to-new task.
>
> > ### **Q4.3 Ablation studies of $\eta$.**
>
> Thank you for your careful reading! As you can observe, the accuracy performance declines sharply when $\eta$ exceeds a certain threshold and keeps increasing. But we want to point out that a small $\eta$ does not indicate the CT regularization negatively impacts performance. The CT regularization term plays its role when $\eta>0$. We would also like to point out that the small $\eta$ is mainly due to the different scales between the ELBO and CT losses. We will add the above discussion in the revision.
>
> Moreover, we have compared PBPrompt with its variant B-Prompt($\eta=0$) at Table 3 and Table4 in the manuscript. We find that B-Prompt outperforms others in most cases, and when we choose a appropriate $\eta$, the performance of PBPrompt ($\eta=0.01$) can be further enhanced. We attribute this to our CT-based alignment, where the generated prompts are encouraged to  closely align with a variety of patch embeddings, and thus preventing them from overfitting to the training data.
> We appreciate your advice, and we will add the case when $\eta=0$ (B-Prompt) in Fig 6 of the revision.
>
> Thanks for your helpful suggestions again! We look forward to having more conversations with you about improving our paper once more.

---

### Official Review · Reviewer_KBov · 2024-03-25

**Q2-1 Originality-Novelty:** 3
**Q2-2 Correctness-Technical Quality:** 4
**Q2-5 Clarity Of Writing:** 4

**Q1 Summary And Contributions:**

The article presents a novel approach to prompt tuning for vision-language models, addressing the challenge of creating effective prompts that can describe diverse characteristics of categories. The motivation stems from the limitations of existing prompt engineering methods, which either require manual designs or optimize prompts as point estimations, potentially leading to overfitting and limited application. Experimental results over 15 datasets show good transferability and generalization performance of our proposed model, both quantitatively and qualitatively.

**Q2-3 Extent To Which Claims Are Supported By Evidence:**

3: Good: the main claims are supported by convincing evidence (in the form of adequate experimental evaluation, proofs, (pseudo-)code, references, assumptions).

**Q2-4 Reproducibility:**

3: Good: key resources (e.g. proofs, code, data) are available and key details (e.g. proofs, experimental setup) are sufficiently well-described for competent researchers to confidently reproduce the main results.

**Q3 Main Strengths:**

Strengths:

1. The whole storyline from motivation to implementation is clear and good. This paper proposed a Bayesian prompt tuning method to generate label-specific stochastic prompts hierarchically. Each label is modeled as a distribution over the embedding space by the introduction of uncertainty.
2. While dealing with semantic regularization to align textual prompts with visual patches, ensuring the prompts capture true label-specific visual concepts, the authors considered the overfitting issues and used CT to prevent overfitting
3. Rigid mathematical backbones

**Q4 Main Weakness:**

Weaknesses:

1.  Lacking full comparison methods. There are a lot of recent methods that may be considered the baseline, e.g., "Prompt-aligned Gradient for Prompt Tuning, ICCV23". And current improvements compared to CoCoop are not as significant (Table 1). This makes the effectiveness of the proposed method in practice not fully convinced.
2. It seems that the most important part is the "Regularization Between Textual Prompts and Visual Patches", which leads to the trade-off between the original classification loss and the diversity of the learned prompt. Currently, experimental results do not fully explore the mechanism behind them.
3. The mathematical annotation is a little bit complicated and confusing.

**Q5 Detailed Comments To The Authors:**

Please check the weaknesses part.

**Q9 Complying With Reviewing Instructions:**

Yes

---

> ### Author Rebuttal · Authors · 2024-04-07
>
> We would like to thank our reviewer KBov for his valuable comments and suggestions, which have improved the quality of our submission. Below, we address the concerns raised in your review.
>
> >  **Q4.1 Lacking full comparison methods and slight improvements CoCoOp in Table 1.**
>
> Thank you for the suggestion, we report the additional results of ProGrad on few-shot learning and cross-dataset transfer learning tasks at Table R1 and R2 respectively. For a fair comparison, We rerun ProGrad with ViT-B/16 and set the prompt length as 4. Compared to ProGrad which only optimizes the prompt whose gradient is aligned to the CLIP knowledge, our approach aims to squeeze CLIP knowledge by finding the stochastic prompts for each class, showing greater potential in capturing diverse visual attributes and improving generalizability. We will add detailed comparisons with ProGrad in the revision.
>
> **Table R1 Comparison with ProGrad on few-shot learning**
> |  Datasets  | Methods | 1 shot| 2 shots  | 4 shots|
> |--|--|--|--|--|
> | Caltech101 | CoOp | 93.13 |  92.97  |  94.50   |
> |    | ProGrad |  93.67  |  94.33   |  94.60   |
> |    | PBPrompt | **93.92** |  **94.40**   | **94.83**    |
> | DTD   | CoOp |  50.03  |  53.94   |  59.23   |
> |    | ProGrad   |  51.12   |   52.30  |  56.00   |
> |    | PBPrompt   |  **52.03**   |  **56.20**   |  **59.63**   |
> | EuroSAT   | CoOp   |  51.80   |  66.33   |  65.87   |
> |    | ProGrad   |  56.65   |   60.65  |  68.70   |
> |    | PBPrompt   |  **60.92**   |  **68.77**   |  **72.84**   |
> | FOOD101  | CoOp   |  82.70   |  82.77   | 86.50 |
> | | ProGrad   |   85.55  |  85.75   |  86.17   |
> | | PBPrompt   |  **85.55**   |   **86.25**  |  **86.30**|
> | SUN397   | CoOp   |  67.32   |  67.67   |  70.14   |
> | | ProGrad   |  67.92   |  68.95   |  70.17   |
> | | PBPrompt |   **68.10** | **69.35** |  **70.21** |
> | UCF101 | CoOp   |  70.07   |  73.30   |  77.87   |
> | | ProGrad |**72.65** |   73.60  | 77.40   |
> | | PBPrompt |71.45|  **74.90**   |  **77.60**   |
>
> **Table R2 Comparison with ProGrad on cross-dataset transfer learning**
> |  Methods  | ImageNet | Caltech101  |  Pets  | Cars  | Flowers |Food|Aircraft|SUN|DTD|EuroSAT|UCF|Average|
> |--|:--:|:--:|:--:|:--:|:--:|:--:|:--:|:--:|:--:|:--:|:--:|:--:|
> |  CoCoOp  | 71.02 | 94.43  |  90.14 | 65.32  | 71.88 | 86.06 |  22.94 |  67.36 |**45.73**| 45.37 |   68.21|  65.74   |
> |  ProGrad | 71.50 | 94.45  | 90.20  | 65.54  |  71.90 | 86.10 |  23.80 |  67.57 | 44.30| 45.70 | **68.90** |  65.85  |
> |  PBPrompt  |**71.71**| **94.87** |**90.62** |**66.00** |**72.44** |**86.34** |**24.82** |**67.69** |45.62 |**47.13** |68.83 |**66.40** |
>
> As for the slight improvements over CoCoOp on the cross-dataset transfer learning task. We would like to point out that:
> 1. The current outstanding works have already achieved high performance in this setting. It is a nontrivial contribution to have consistent improvement over 11/12 datasets at Table 1. From Table. R2, we find that both ProGrad and our PBPrompt achieve marginal results over CoCoOp. Our approach beats ProGrad in most cases, which demonstrate the efficiency of the proposed model.
>
> 2. Besides the cross-dataset transfer learning, our method achieves significant improvements on other tasks. For example, PBPrompt achieves 0.9%-9.14 % improvements compared to CoCoOp in terms of **H** score (Figure 4 in the manuscript and detailed results in Appendix.).
>
> 3. Beyond the numerical comparisons, The visualization in Figure 7 shows the interpretability of the learned prompts, while CoCoOp fails to provide such visualization technically, due to their non-interpretable framework.
>
> >  **Q4.2 It seems that the most important part is the…**
>
> One of the core ideas behind PBPrompt is to learn stochastic prompts to search diverse visual concepts for each class. The CT-based regularization is built upon the stochastic prompts generation (SPG) to align the sampled textual prompts with the visual patches, and guide the learned prompts to capture the true class-specific visual attributes rather than overfitting to the training patterns. To test the CT-based regularization impact empirically, we compare PBPrompt with its variant B-Prompt (B-Prompt removes the regularization and only contains the SPG module) on the few-shot and base-to-new tasks (Table 3 and 4 in the manuscript). We find that PBPrompt achieves higher scores in 9/12 cases on the base-to-new task and 22/24 cases on the 1/2/4 shots task, which empirically demonstrates the effectiveness of the CT-based regularization. Besides, the visualization at Fig 7 and Fig D2 in the manuscript also prove the alignments across modalities.
>
> >  **Q4.3 The mathematical annotation is a little bit complicated and confusing.**
>
> Thank you for your careful reading! We will check all the formulas and simplify their mathematical annotations in the revision.
>
> Thanks for your helpful suggestions again! We look forward to having more conversations with you about improving our paper once more.

---

### Meta-Review · Area_Chair_Hcg6 · 2024-04-15

The paper introduces a Bayesian prompt tuning method for vision-language models, such as CLIP. The generated prompts are class-specific and stochastic, sampling from a learned distribution. Unlike typical deterministic methods, the stochastic nature of these prompts is able to capture a diverse range of category characteristics that deterministic prompts often miss. The main concern from the reviewers is the incremental improvement in empirical results. In response, the authors provided additional experimental results to further demonstrate the improvement.
All reviewers agree that this paper is above the acceptance threshold.